# GRAPH REWIRING BASED ON FLOW ALIGNMENT FOR IMPROVING FLUID SIMULATION

## ABSTRACT

To overcome computation burden of traditional computational fluid dynamics (CFD) simulations, researchers have explored different architectures to develop physics-informed simulation methods. Among them, graph neural networks (GNN) are most suitable for adopting CFD meshes, which are extensively used in engineering and industrial applications. However, classical GNNs propagate information among neighbour nodes, which highly restrict information exchange within the network. To address this issue, graph rewiring methods have been developed for generic graph problems, but not particular for fluid simulation. PIORF, introducing edges connecting distant nodes, is the first graph rewiring method to do so, and previous experiments have demonstrated its effectiveness against state-of-the-art generic rewiring methods. Nevertheless, in this work, we found that simply connecting all 2-hop nodes can provide competitive performance with PIORF. This result raises three questions: 1) Is physics-informed rewiring really useful for improving flow predictions? 2) Should we consider just local connection, instead of connecting distant nodes? 3) Do we need to change the connections based on input flow for rollout simulations? By thoroughly adopting physical fluid principles, we propose a simple yet very efficient method, Flow Alignment Rewiring (FLARE) technique, which connects 2-hop nodes only when the node direction aligns with input flow direction. Hence, FLARE is a physics-informed local rewiring method, different from PIORF and well-aligned with fluid physics. Extensive numerical experiments on flows over a cylinder and single and tandem airfoil under different flow conditions and deep network architectures demonstrate that FLARE outperforms PIORF and various 2-hop rewiring approaches by a significant margin.

## 1 INTRODUCTION

Computational fluid dynamics (CFD) is widely employed in engineering to simulate fluid flows around objects. Traditionally, CFD involves solving the Navier–Stokes equations numerically, requiring sufficiently fine meshes, especially near boundary layers and wakes, where flow behaviors vary rapidly (Rumsey & Ying, 2002; Spalart & Venkatakrishnan, 2016). Although mesh refinement substantially improves simulation accuracy, it significantly increases computational demands, often rendering high-fidelity simulations resource intensive. To address the limitations of traditional numerical solvers, deep learning methods with integration of physics prior knowledge have been investigated and considered as a viable solution. Physics-informed neural networks (PINNs), introduced by Raissi et al. (2019), integrate physical equations directly into neural network training, laying the foundational work for future physics-informed machine learning methods. Although PINNs showed early success, they often encountered scalability and generalization challenges in complex fluid scenarios involving multiple interacting features (Krishnapriyan et al., 2021; Wang et al., 2022).

Various common neural architectures such as multilayer perceptron (MLP), convolutional neural network (CNN), and graph neural network (GNN) have been employed in previous fluid simulation studies (Raissi et al., 2019; Tompson et al., 2017; Sanchez-Gonzalez et al., 2020; Pfaff et al., 2020). Among them, GNNs attract considerable attention for their abilities to take CFD mesh as an input directly and exploit the prior knowledge in the mesh, e.g., a dense cell region corresponding to a region with fast-changing velocity and/or pressure. However, this direct adoption of CFD mesh in

GNN inherits the weaknesses from both GNN and the mesh. More precisely: 1) Classical GNN only propagates information to connected neighbors, thus limiting its speed of information exchange. 2) The mesh constructed for discretizing the governing equations is generally not related to flow direction. 3) In a rollout simulation with static object(s), even though velocities of different regions may change constantly, the same graph structure, and hence the mesh, is used in the entire simulation.

The first weakness is well-known in the AI community (Alon & Yahav, 2020) so various rewiring methods have been proposed (Micheli & Tortorella, 2025). These non-physics-informed rewiring methods are designed to identify information bottleneck nodes through the topology of the graph and distribute the information to less information-congressed regions. Since they are developed for generic graph problems, no physical quantities are required in their rewiring. Yu et al. (2025) pinpointed the state-of-the-art generic rewiring methods, including DIGL (Gasteiger et al., 2019), SDRF (Topping et al., 2021), FoSR (Karhadkar et al., 2022) and BORF (Nguyen et al., 2023), are ineffective for fluid simulation. Their results imply that there are some fundamental differences between fluid simulation and graph problems studied in previous works.

To improve fluid simulation, Yu et al. (2025) proposed a new rewiring method, PIORF, which uses Ollivier–Ricci curvature (ORC) to identify bottleneck nodes and connects those nodes to high-velocity gradient nodes. Like other methods, PIORF measures information congression based on topology of the graph but its connections based on velocity gradient were not employed by any previous methods. PIORF allows long distant nodes to be connected, and all connections are bidirectional, as illustrated in Fig. 1(a). It achieves optimal performance when ORC selects $3\% - 7\%$ of nodes (dataset-dependent) with the highest information compression for rewiring. Selecting more nodes would degrade its performance. PIORF can consistently achieve improvements against baselines with and without state-of-the-art rewiring methods.

However, PIORF does not totally align with physical principles. First, contrary to PIORF's distant connections, fluid flow convects and diffuses locally. Also, net fluid flux across space is not bidirectional. Because of these misalignments with physical principles yet good performance, PIORF draws our attention. We compare PIORF with all 2-hop connections, which add bidirectional edges to all nodes with 2-hop distances, as illustrated in Fig. 1(b). It is a local and non-directional[1] connection scheme. Surprisingly, this simple and non-physics-informed method can provide competitive performance against PIORF. For example, the average velocity RMSE of the PIORF and the 2-hop connection method (2-HOP-ALL in Tab. 2) over three baseline architectures on the *CylinderFlow* database (Pfaff et al., 2020) are 55.23 and 46.97, and their average velocity RMSE on the *Airfoil* dataset (Pfaff et al., 2020) are 38.69 and 39.47, respectively. More details about this comparison are given in the experiment section. Although we should mention that PIORF has lower training cost compared with the 2-hop connection, these experimental results raise three important questions:

1. Is physics-informed based rewiring really useful for improving fluid flow simulations?

2. Should we consider just local connection, instead of connecting distant nodes?

3. Do we need to change the connections based on input flow for rollout simulations?

To answer these questions and develop an effective rewiring method for fluid simulation, we propose the Flow Alignment Rewiring (FLARE) method, which is designed based on physics principles, considering local and input flow direction for connections, as shown in Fig. 1(c). The contributions of this work include:

- FLARE is the first physics-informed rewiring method aligned with physical principles[2].

- FLARE is the first rewiring method using local information on flow velocity and direction for directional connection.

- Extensive experiments on the *CylinderFlow*, *Airfoil*, and *Tandem-Airfoil-Cruise* datasets with different flow conditions demonstrate that FLARE outperforms PIORF, baselines, and variations of 2-hop connections, indicating its effectiveness and robustness.

---

[1]Because of the bidirectional connections, information from node $i$ can flow to node $j$ and vice versa, same as the original graph constructed from CFD Mesh, so the scheme is considered non-directional.

[2]PIORF does use the term physics-informed and velocity gradient to determine its connections, but it does not align with physical principles as explained in the introduction.

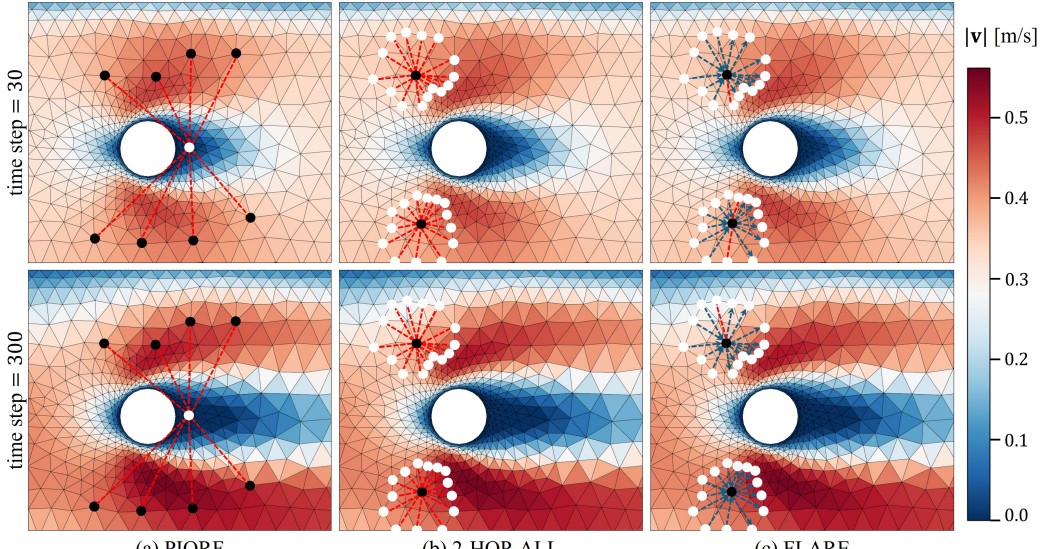

Figure 1: Illustration of different rewiring (dashed-line) under different connections: (a) PIORF's rewiring by connecting the bottleneck nodes (black) to the largest velocity difference node (white), (b) 2-HOP-ALL connection that satisfies Eqn. (2), and (c) FLARE, the proposed rewiring connections that are local, directional, and based on current flow directions as labeled in blue arrows, whereas the existing bidirectional edges (red dashed-line) are due to opposite flow directions between two corresponding nodes.

## 2  RELATED WORK

Recent years have seen the rapid adoption of graph neural networks (GNNs) for simulating unsteady fluid flows, owing to their flexibility on irregular meshes and strong inductive biases for physical interactions. Pioneering works (Sanchez-Gonzalez et al., 2020; Pfaff et al., 2020) have demonstrated the potential of message-passing frameworks to capture dynamics across fluids, rigid bodies, and deformable solids on unstructured domains. Subsequently, other models (Obiols-Sales et al., 2020; Chen et al., 2021; Jessica et al., 2023; Lim et al., 2025) applied similar architectures to predict velocity and pressure fields efficiently, achieving significant acceleration compared to traditional CFD solvers. More recent extensions addressed scalability and long-range dependencies. X-MeshGraphNet (Nabian et al., 2024) introduced domain partitioning and multi-scale halo exchange to improve scalability, while AMGNet (Yang et al., 2022) and BSMS-GNN (Cao et al., 2023) incorporated multi-scale pooling for efficient simulation on large meshes. These models consistently demonstrated a speedup of several orders of magnitude while retaining accuracy across laminar and turbulent regimes. To further increase fidelity, physics-informed GNNs (Chen et al., 2021; Belbute-Peres et al., 2020; Lim et al., 2024; Yu et al., 2024) have been explored through integration of governing equations as soft constraints or coupling with a differentiable PDE solver. These advances highlight GNNs as promising surrogates for high-dimensional, unsteady fluid simulations, though they remain challenged by long-range information propagation and bottleneck issues deeply tied to over-squashing.

Early theoretical studies characterized over-squashing as information contraction (Banerjee et al., 2022) and vanishing sensitivity in deep GNNs (Di Giovanni et al., 2023), while also uncovering trade-offs with over-smoothing mediated by the spectral gap (Giraldo et al., 2023). Geometry has played a key role. For instance, Topping et al. (2021) introduced curvature-based rewiring via Stochastic Discrete Ricci Flow, showing that negatively curved edges induce over-squashing. Nguyen et al. (2023) extended this with Batch Ollivier–Ricci Flow (BORF), unifying over-squashing and over-smoothing via local curvature. These previous works are developed for generic graph learning problems, not specifically for fluid simulations. They rely on the graph topology to determine the connections, and no physical quantities, such as flow velocity, are involved in the determination.

Based on previous works, PIORF (Yu et al., 2025) leverages Ollivier–Ricci flow to identify information bottleneck nodes and determines their connections with other nodes using velocity gradients to enhance long-range interactions in mesh-based GNNs. It is the first work to exploit velocity gradients in rewiring, allowing long-distance, bidirectional connections. However, as mentioned above, such treatments do not align with physical principles. To understand PIORF's performance more deeply and the importance of fluid principles in rewiring, we systemically compare PIORF with the local and non-directional 2-hop connection scheme and develop FLARE based on the principles.

## 3 FLOW ALIGNMENT REWIRING

### 3.1 NOTATIONS AND PRELIMINARIES

**Graph representation:** We represent the computational mesh as a directed graph $G = (V, E)$. Each node $i \in V$ corresponds to a mesh point with position $\mathbf{x}_i$ and input velocity $\mathbf{v}_i$, where $\mathbf{x}_i = [x_i, y_i]^T \in \mathbb{R}^2$ and $\mathbf{v}_i = [u_i, v_i]^T \in \mathbb{R}^2$. Bold symbols, e.g., $\mathbf{x}_i$ and $\mathbf{v}_i$ represent vectors and non-bold symbols, e.g., $x_i$ and $y_i$ represent scalars. Nodes are connected by directional edges, $(i, j) \in E$, where messages are sent from node $i$ (sender) to node $j$ (receiver). For bidirectional connection, edges from node $i$ to node $j$, i.e., $(i, j)$ and from node $j$ to node $i$ i.e., $(j, i)$ are needed. In standard physics-informed AI studies with CFD mesh as the input, all edges are bidirectional, i.e., both $(i, j)$ and $(j, i)$ exist in the graph because the CFD mesh is non-directional. The graph of the previous rewiring work PIORF is also bidirectional. Different from previous studies, our graph is directional, meaning that the presence of edge $(i, j)$ in the graph does not imply that edge $(j, i)$ also exists.

**Message passing:** Let $h_i^{(l)}$ denote the hidden state of node $i$ at layer $l$ of a GNN, and $e_{ij}$ be the edge feature on $(i, j)$. A generic message passing layer is represented as:

$$\mathbf{m}_{i \to j}^{(l)} = m_\phi(\mathbf{h}_i^{(l)}, \mathbf{h}_j^{(l)}, e_{ij}), \quad \mathbf{h}_j^{(l+1)} = u_\theta \left( \mathbf{h}_j^{(l)}, \sum_{i:(i,j) \in E} \mathbf{m}_{i \to j}^{(l)} \right), \quad (1)$$

where $m_\phi$ and $u_\theta$ are network components for updating the hidden state through the features on connected edges and nodes. In the rollout simulation, $h_i^{(0)}$ representing input features can include output velocity and density of the previous timestep and other features such as signed distance function (SDF) and directional integration distance (DID) (Jessica et al., 2023). In static simulation, input feature can also include SDF, DID and preliminary velocity estimates from another method (Jessica, 2025).

**2-hop connection:** The proposed rewiring method FLARE considers local connections instead of long-distance connection used in PIORF. 2-hop connection is the shortest connection, except for those 1-hop connections in the original graph from CFD mesh. A pair of nodes $(i, j)$ is 2-hop connected if there exists a node $k \in V$ with $(i, k) \in E$ and $(k, j) \in E$, while the direct edge $(i, j) \notin E$. The set of all 2-hop connected edges is defined as:

$$C_2 = \{(i, j) \in V \times V : \exists k \in V, (i, k) \in E, (k, j) \in E, (i, j) \notin E, i \neq j\}. \quad (2)$$

### 3.2 PHYSICAL PRINCIPLES AND FLARE

The proposed FLARE method is developed based on fundamental physical principles of fluid dynamics. First, at any given time, the net fluid transport between two neighboring regions is inherently unidirectional. In other words, fluid mass flows from one region to another without simultaneous reverse transport. Second, fluid mass is a physical quantity constrained by locality, for which it can only propagate over short spatial distances, $\Delta \mathbf{x}$, within a finite time interval, $\Delta t$, and cannot instantaneously appear in distant regions. These two principles form the basis of classical numerical solvers, such as the finite volume method (Moukalled et al., 2016), which are widely employed in engineering and industrial fluid simulations. Lastly, the transport of fluid mass is determined by the instantaneous velocity field, $\mathbf{v}$, meaning that movement is aligned with the current prevailing flow direction. Figure 2 illustrates these three principles: (i) unidirectionality, (ii) locality, and (iii) flow alignment that establish the concept of FLARE.

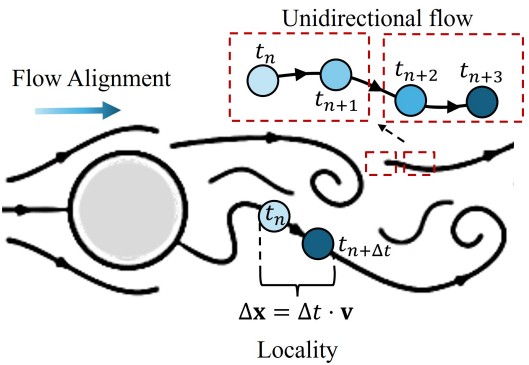

Figure 2: Illustration of the three physical principles underlying the proposed FLARE method in the context of flow over a cylinder: (i) unidirectional flow, fluid transport occurs in a single direction between regions, (ii) locality, transport is constrained to short spatial distance proportional to $\Delta x = \Delta t \cdot \mathbf{v}$, and (iii) flow alignment, connections follow the instantaneous flow direction. The progressively darker circles indicate temporal progression of fluid elements from $t_n$ to $t_{n+3}$ or $t_{n+\Delta t}$.

Adopting the three principles in the development of a rewiring method, the rewiring connections should be local, directional, and based on current flow direction, which are respectively suggested by the first, second, and third principles. Although PIORF also uses velocity in their rewiring and claims to be a physical-informed method, it violates all three principles. From the perspective of fluid research, it is therefore physically invalid. In contrast, the proposed FLARE is designed based on the three principles. FLARE only considers directional 2-hop connections, which are the most local connections, except those connections in the original CFD mesh, and its connections are determined by the velocity of input flow. Given each candidate $(i, j) \in C_2$, their displacement vector $\mathbf{d}_{ij}$ can be obtained by:

$$\mathbf{d}_{ij} = \mathbf{x}_j - \mathbf{x}_i = [x_j - x_i, y_j - y_i]^T. \tag{3}$$

The flow alignment score $s_{ij}$ is defined by the projection of the velocity vector of the sender onto the displacement vector:

$$s_{ij} = \mathbf{v}_i^T \mathbf{d}_{ij} = u_i(x_j - x_i) + v_i(y_j - y_i). \tag{4}$$

It is worthy to highlight that the flow alignment score only uses velocity of the sender, different from PIORF, which uses velocity of both nodes to compute velocity gradient. FLARE selects 2-hop connections based on flow alignment for rewiring. More precisely, FLARE connects nodes $(i, j) \in C_2$ only when $s_{ij} > T$, a predefined threshold. The selected connections form the set:

$$A_{align} = \{(i, j) \in C_2 : s_{i,j} > T\}. \tag{5}$$

The connections in $A_{align}$ and the original graph $G = (V, E)$ from the mesh form a new graph as:

$$G^+ = (V, E \cup A_{align}). \tag{6}$$

This new rewired graph fulfills the three principles. Note that in rollout simulation, this graph keeps changing because flow velocity field at each point of time is different, as shown in Fig. 1(c).

During training, the ground truth velocity field at time $t$ is used to derive $A_{align}$ and $G^+$ to predict flows at time $t + 1$. At time of inference, a velocity field at time 0 given by another method is used in the initial rewiring. If no velocity field is given at the beginning, FLARE will not rewire the graph at time 0. It will rewire graph connections in the rest of the time steps based on the output flow of the previous time steps. Following PIORF's setting, in our experiments, we use ground truth velocity field to rewire the graph at the starting time point. By using 2-hop connections, FLARE can extend information propagation range from $L$ to $2L$ for classical GNN, where $L$ is the total number of layers in the network.

## 4 EXPERIMENTS

To evaluate the proposed physics-informed rewiring method FLARE, we systemically compare it with PIORF and various wo 2-hop connections on three datasets with compressible and incompressible steady and unsteady flows.

### 4.1 DATASETS

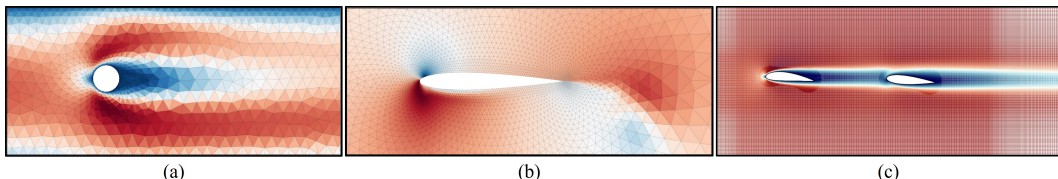

Figure 3: Velocity contours from datasets (a) *CylinderFlow*, (b) *Airfoil*, and (c) *Tandem-Airfoil-Cruise*.

The experiments are conducted on three benchmark datasets: *CylinderFlow* (Pfaff et al., 2020), *Airfoil* (Pfaff et al., 2020), and *Tandem-Airfoil-Cruise* (Jessica, 2025), as illustrated in Fig. 3. *CylinderFlow* and *Airfoil* are widely employed in physics-informed AI studies, including evaluation of PIORF, and thus serve as our primary unsteady flow benchmarks. To further assess performance under steady flow conditions, we additionally employ the *Cruise* subset of the recently generated *Tandem-Airfoil* dataset, which captures complex steady interactions between tandem airfoils.

*CylinderFlow* contains 1200 incompressible unsteady flow simulations around cylindrical obstacles of varying radii. Each simulation consists of 600 time steps with a fixed mesh topology, averaging 1885 nodes per mesh. The dataset is split into 1000 training, 100 validation, and 100 testing simulations. During inference, the ground-truth velocity field at the initial time step is used for rewiring, and subsequent steps rely on the previous time step of model's predicted velocity field.

*Airfoil* comprises 1200 compressible unsteady flow simulations over airfoil geometries. Similar to *CylinderFlow*, each simulation contains 600 time steps with static meshes averaging 5223 nodes. The dataset is partitioned into 1000 training, 100 validation, and 100 testing simulations. As in *CylinderFlow*, graph rewiring at the first time step uses ground-truth velocity, with later steps updated using predicted velocities.

*Tandem-Airfoil-Cruise* dataset consists of 784 incompressible steady flow simulations of tandem-airfoil configurations with average 351315 nodes per simulation. Two airfoils of varying shapes and sizes are randomly sampled and positioned within a bounded region at a fixed angle of attack of $5°$ and Reynolds number of 500. This setup creates complex flow interactions. The dataset is divided into training, validation, and test sets in an 8:1:1 ratio. Following Jessica (2025), initial tandem-airfoil flow fields are estimated by a deep network trained on single-airfoil data, which are then used for graph rewiring in our experiments.

Table 1 summarizes the edge features, node features, and prediction targets for each dataset. Since *CylinderFlow* is incompressible flow, pressure field can be directly derived from velocity via Poisson's equation, and thus we do not include pressure field as our prediction target. For *Airfoil*, we follow the recently released BSMS-GNN implementation to predict the spatial gradients of the velocity and density fields. Additional features, such as signed distance functions employed in Mesh-GraphNets, are incorporated where relevant, ensuring consistency with prior baselines. More details are given in the Appendix.

### 4.2 BASELINES AND EXPERIMENTAL SETTINGS

Since Yu et al. (2025) systemically compared PIORF with the state-of-the-art generic rewiring methods, including DIGL (Gasteiger et al., 2019), SDRF (Topping et al., 2021), FoSR (Karhadkar et al., 2022) and BORF (Nguyen et al., 2023) and concluded that PIORF consistently outperforms them in fluid simulations, we do not include them in this evaluation. Instead, we focus our comparisons on PIORF and 2-hop connections, which can be considered as variations of the proposed FLARE. PIORF serves as the key method in this evaluation as it also uses velocity in its rewiring and was tested

Table 1: Feature specification employed in experiments, average number of nodes, and flow types for the three datasets. Variables denote: $(u_i, v_i)$–velocity components in x and y directions; $\rho_i$–fluid density; $p_i$–pressure; $\nabla(\cdot)$–spatial gradient; $\mathbf{n}_i$–node type; $\mathbf{x}_{ij}$–relative position vector.

| Dataset | Edge Features | Node Features | Prediction Targets | Ave. Nodes | Flow Type |
|---|---|---|---|---|---|
| CylinderFlow | $\mathbf{x}_{ij}, \|\mathbf{x}_{ij}\|$ | $\mathbf{n}_i, u_i, v_i$ | $\nabla u_i, \nabla v_i$ | 1885 | Incompressible, unsteady |
| Airfoil | $\mathbf{x}_{ij}, \|\mathbf{x}_{ij}\|$ | $\mathbf{n}_i, u_i, v_i, \rho_i$ | $\nabla u_i, \nabla v_i, \nabla \rho_i$ | 5223 | Compressible, unsteady |
| Tandem-Airfoil-Cruise | $\mathbf{x}_{ij}, \|\mathbf{x}_{ij}\|$ | $\mathbf{n}_i, u_i, v_i, p_i$ | $u_i, v_i, p_i$ | 351315 | Incompressible, steady |

on fluid simulations. In addition to PIORF, we also include a full 2-hop connection scheme, denoted as 2-HOP-ALL, which uses bidirectional edges to connect all 2-hop nodes directly. 2-HOP-ALL serves as a local and non-physical-informed baseline. Because of the bidirectional connections, 2-HOP-ALL can be regarded as a non-directional scheme, same as PIORF. Since FLARE selectively connects 2-hop nodes based on the physical principles, comparing FLARE with 2-HOP-ALL is an important indicator to validate the necessity of the physical principles. Moreover, a random 2-hop scheme, denoted as 2-HOP-RANDOM, which randomly connects 2-hop nodes with same number of edges as FLARE, is also included. Its performance difference with FLARE is used to evaluate the effectiveness of FLARE's connections based on input flow directions.

In the experiments, MeshGraphNets (MGN) (Pfaff et al., 2020), BSMS-GNN (Cao et al., 2023) and Transolver++ (Luo et al., 2025) are employed as baseline architectures. The first two are graph networks and adopted by PIORF's study. For MeshGraphNets, we strictly adhere to established experimental configurations and dataset splits (Pfaff et al., 2020), ensuring comparability by modifying only edge connectivity without altering existing message transmission or update mechanisms. In the BSMS-GNN (Cao et al., 2023; Yu et al., 2025) setting, the rewiring methods specifically applied at the finest resolution level. It is worth noting that BSMS-GNN has a hierarchical scheme to improve message passing. Transolver++ is a transformer-based architecture. We include it in our evaluation to test FLARE on transformer-based architecture, which is not FLARE designed for. Given that Transolver++ does not have explicit edge-based messaging, we integrate FLARE through adding message-passing (MP) blocks on top of the transformer blocks, facilitating effective utilization of the rewired graph structure. We validate this by prepending two MP blocks to Transolver++ and compare them on *CylinderFlow*. The standard Transolver++ achieves RMSE of $38.12 \times 10^{-3}$ and the revised Transolver++ with additional 2MP blocks achieves RMSE of $32.77 \times 10^{-3}$. The two MP blocks provide over 14% improvement on RMSE. Thus, we employ the revised Transolver++ to evaluate different rewiring methods.

All experiments and implementation are conducted using PyTorch, leveraging publicly available codebases to ensure reproducibility and transparency. Experiments are executed on NVIDIA RTX 5090 GPUs with W9-3475X CPUs, and complete training details, hyperparameters, and supplementary evaluation specifics are comprehensively documented in the appendix to facilitate reproducibility and future research. Upon acceptance, we will share our codebases.

### 4.3 RESULTS

Table 2 shows the full rollout RMSE on *CylinderFlow* and *Airfoil*. For *CylinderFlow*, we can see that 2-HOP-ALL consistently outperforms PIORF and the baseline models without using any rewiring schemes. For *Airfoil*, 2-HOP-ALL performs very similar to PIORF in most of the comparisons, except for density predictions of MGN and BSMS-GNN, $95.04 \times 10^{-3}$ vs $86.49 \times 10^{-3}$ and $99.75 \times 10^{-3}$ vs $128.96 \times 10^{-3}$. These results indicate that in terms of accuracy, PIORF has no clear advantages over the non-physics-informed scheme, 2-HOP-ALL. Comparing FLARE with PIORF and 2-HOP-ALL on *CylinderFlow*, we can observe clear performance gains from FLARE. On average, FLARE achieves 27.40% improvement over PIORF and 26.03% improvement over 2-HOP-ALL.

Figure 4 shows the RMSE of MGN with different rewiring methods in rollout simulation. The RMSE of all rewiring methods have the same trend. After the 20th time step, RMSE of all methods increase. However, RMSE of FLARE increases significantly slower than the others. It is worth noting that PIORF performs similarly as 2-HOP-RANDON in this experiment.

Table 2: Full-rollout RMSE on *CylinderFlow* and *Airfoil* datasets (mean $\pm$ SE). Values scaled by $\times 10^3$ except *Airfoil* velocity. Best results are highlighted in **bold**, second-best results are underlined.

| Model | Method | *CylinderFlow* | | *Airfoil* **Velocity** | | *Airfoil* **Density** | |
|---|---|---|---|---|---|---|---|
| | | RMSE ($\times 10^3$) | Improv. | RMSE | Improv. | RMSE ($\times 10^3$) | Improv. |
| **MGN** | Baseline | $40.35 \pm 4.30$ | - | $35.45 \pm 2.33$ | - | $94.39 \pm 6.28$ | - |
| | PIORF | $33.59 \pm 3.70$ | 16.8% | $33.66 \pm 2.17$ | 5.1% | $95.04 \pm 5.79$ | -0.7% |
| | 2-HOP-ALL | $\underline{29.40 \pm 2.60}$ | $\underline{27.1\%}$ | $34.03 \pm 2.62$ | 4.0% | $86.49 \pm 5.71$ | $\underline{8.4\%}$ |
| | 2-HOP-RANDOM | $33.52 \pm 3.20$ | 16.9% | $33.33 \pm 2.28$ | 6.0% | $95.16 \pm 5.85$ | -0.8% |
| | FLARE (ours) | $\mathbf{23.38 \pm 2.50}$ | $\mathbf{42.1\%}$ | $\underline{33.27 \pm 2.39}$ | $\underline{6.1\%}$ | $90.58 \pm 5.76$ | 4.0% |
| | FLARE + 10% Density 2HOP | - | - | $\mathbf{31.93 \pm 2.57}$ | $\mathbf{9.9\%}$ | $\mathbf{85.66 \pm 6.12}$ | $\mathbf{9.2\%}$ |
| **BSMS-GNN** | Baseline | $97.15 \pm 6.80$ | - | $46.57 \pm 3.20$ | - | $126.78 \pm 8.14$ | - |
| | PIORF | $101.01 \pm 5.70$ | -4.0% | $44.25 \pm 2.80$ | 5.0% | $\underline{99.75 \pm 6.93}$ | $\underline{21.3\%}$ |
| | 2-HOP-ALL | $79.60 \pm 5.32$ | 18.1% | $45.91 \pm 3.10$ | 1.4% | $128.96 \pm 8.12$ | -1.7% |
| | 2-HOP-RANDOM | $\underline{64.80 \pm 3.73}$ | $\underline{33.3\%}$ | $51.84 \pm 3.10$ | -11.3% | $195.92 \pm 6.36$ | -54.6% |
| | FLARE (ours) | $\mathbf{56.28 \pm 3.80}$ | $\mathbf{42.1\%}$ | $\underline{43.63 \pm 3.31}$ | $\underline{6.3\%}$ | $110.85 \pm 9.30$ | 12.6% |
| | FLARE + 10% Density 2HOP | - | - | $\mathbf{39.46 \pm 2.90}$ | $\mathbf{15.3\%}$ | $\mathbf{95.58 \pm 7.05}$ | $\mathbf{24.6\%}$ |
| **Transolver++** | Baseline | $32.77 \pm 4.09$ | - | $40.27 \pm 2.19$ | - | $73.76 \pm 4.61$ | - |
| | PIORF | $\underline{31.10 \pm 3.74}$ | $\underline{5.1\%}$ | $38.16 \pm 2.78$ | 5.2% | $73.89 \pm 5.04$ | -0.2% |
| | 2-HOP-ALL | $31.90 \pm 2.96$ | 2.7% | $38.49 \pm 2.52$ | 4.4% | $74.66 \pm 4.97$ | -1.2% |
| | 2-HOP-RANDOM | $32.21 \pm 3.19$ | 1.7% | $37.36 \pm 2.80$ | 7.2% | $73.89 \pm 5.04$ | -0.2% |
| | FLARE (ours) | $\mathbf{28.76 \pm 3.16}$ | $\mathbf{12.2\%}$ | $\underline{35.40 \pm 2.40}$ | $\underline{12.1\%}$ | $\underline{67.93 \pm 4.48}$ | $\underline{7.9\%}$ |
| | FLARE + 10% Density 2HOP | - | - | $\mathbf{34.26 \pm 2.32}$ | $\mathbf{14.9\%}$ | $\mathbf{64.76 \pm 4.71}$ | $\mathbf{12.2\%}$ |

For velocity field of *Airfoil*, among the basic compared methods (Baseline, PIORF, 2-HOP-ALL, and 2-HOP-RANDOM), FLARE always performs the best. It achieves 3.26% improvement over PIORF and 5.38% improvement over 2-HOP-ALL. For density field of *Airfoil*, FLARE performs either the best or the second best. On average error over the three models, FLARE and PIORF perform very similarly for the density prediction. Comparing with 2-HOP-ALL, FLARE provides 6% improvement for density prediction. It is worth noting that FLARE gains more significant improvements on *CylinderFlow* than *Airfoil*, we performed an analysis, whose details are given in Appendix A.5. Comparing with 2-HOP-RANDOM, FLARE outperforms it in all comparisons. In addition, Table 2 includes an extended variant with separate rewiring for density ("FLARE + 10% Density 2HOP"), which further improves both velocity and density on *Airfoil* across all three models; the details of this extension is introduced in Appendix A.4.

Table 3 shows the results of *Tandem-Airfoil-Cruise*. We do not include BSMS-GNN in this comparison because of its memory requirements exceeding our equipment limits and its lowest performance on *CylinderFlow* and *Airfoil*. On average, FLARE provides 14.43% performance gain over PIORF and 2.50% performance gain over 2-HOP-ALL. As with results in Table 2, FLARE consistently outperforms 2-HOP-RANDOM.

The experimental results on the three datasets indicate that FLARE outperforms 2-HOP-ALL and PIORF. These results demonstrate the effectiveness of our design combining flow alignment, directionality, and 2-hop locality. FLARE achieves consistent improvements across different flow dynamics regimes in the three datasets, with particularly strong benefits in regions with evolving flow patterns.

Table 3: MSE on *Tandem-Airfoil-Cruise* dataset (Reynolds number, $\mathrm{Re} = 500$; AoA, $\alpha = 5°$). Mean $\pm$ SD values are scaled by $\times 10^3$.

| Model | Method | MSE ($\times 10^3$) | Improv. |
|---|---|---|---|
| **MGN** | Baseline | $67.53 \pm 33.82$ | - |
| | PIORF | $18.95 \pm 26.61$ | 71.9% |
| | 2-HOP-ALL | $\underline{12.76 \pm 15.94}$ | $\underline{81.1\%}$ |
| | 2-HOP-RANDOM | $24.57 \pm 29.29$ | 63.6% |
| | FLARE (ours) | $\mathbf{11.68 \pm 11.47}$ | $\mathbf{82.7\%}$ |
| **Transolver++** | Baseline | $1.04 \pm 0.76$ | - |
| | PIORF | $0.71 \pm 0.71$ | 31.7% |
| | 2-HOP-ALL | $\mathbf{0.66 \pm 0.67}$ | $\mathbf{36.5\%}$ |
| | 2-HOP-RANDOM | $0.86 \pm 0.71$ | 17.3% |
| | FLARE (ours) | $\underline{0.68 \pm 0.67}$ | $\underline{34.6\%}$ |

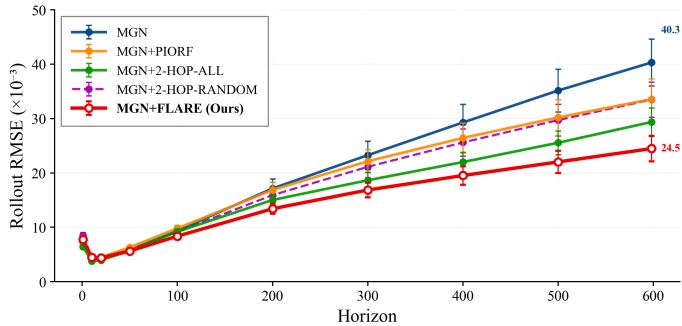

Figure 4: Rollout prediction error on *CylinderFlow* test set. Error bars indicate standard error across 100 trajectories. The initial decrease in error reflects models' robustness to noise levels encountered during training, while subsequent increase occurs as accumulated prediction errors exceed training noise magnitudes. FLARE achieves the lowest long-horizon error.

## 4.4 ABLATION STUDY

In the first ablation experiment, we study the importance of connection direction. We inversely connect the FLARE to form a new rewiring scheme denoted as 2-HOP-OPPOSITE. Table 4(a) clearly indicates that the inverse connections deteriorate the performance on both *CylinderFlow* and *Airfoil* datasets. Comparing results in Table 2, we can observe that 2-HOP-OPPOSITE performs even worse than 2-HOP-RANDOM. These results demonstrate that the selection of connection directions has significant impact on prediction performance.

In the second ablation experiment, we study the effect of applying flow alignment rewiring on longer hop distances. We extend FLARE to select 3-hop or 4-hop connections based on flow alignment, while applying adaptive thresholds to maintain connection counts aligned with basic FLARE, denoted as FLARE-3HOP and FLARE-4HOP respectively. Table 4(b) clearly indicates that FLARE-3HOP and FLARE-4HOP underperform standard FLARE on *CylinderFlow*. This further confirms the importance of local connections guided by the physical principles. These results suggest that 2-hop connections balance between local flow alignment and computational efficiency, with longer-range connections potentially introducing noise that degrades prediction stability.

Table 4: Ablation studies on rewiring strategies. Full-rollout RMSE values with *CylinderFlow* velocity and *Airfoil* density scaled by $\times 10^3$.

**(a) Flow Direction Comparison**

| Model | Method | *CylinderFlow* | *Airfoil* | |
|---|---|---|---|---|
| | | Velocity($\times 10^3$) | Velocity | Density($\times 10^3$) |
| MGN | 2-HOP-OPPOSITE | $55.48 \pm 3.10$ | $34.17 \pm 2.36$ | $96.39 \pm 5.50$ |
| | FLARE (ours) | $\mathbf{23.38 \pm 2.50}$ | $\mathbf{33.27 \pm 2.39}$ | $\mathbf{90.58 \pm 5.76}$ |
| BSMS | 2-HOP-OPPOSITE | $69.96 \pm 4.50$ | $47.77 \pm 2.95$ | $162.31 \pm 7.23$ |
| | FLARE (ours) | $\mathbf{56.28 \pm 3.80}$ | $\mathbf{43.63 \pm 3.31}$ | $\mathbf{110.85 \pm 9.30}$ |
| Trans++ | 2-HOP-OPPOSITE | $29.35 \pm 3.05$ | $40.85 \pm 2.63$ | $72.20 \pm 5.07$ |
| | FLARE (ours) | $\mathbf{28.76 \pm 3.16}$ | $\mathbf{35.40 \pm 2.40}$ | $\mathbf{67.93 \pm 4.48}$ |

**(b) Hop Distance Impact**

| Method | *CylinderFlow* Velocity($\times 10^3$) |
|---|---|
| FLARE-2HOP | $\mathbf{23.38 \pm 2.50}$ |
| FLARE-3HOP | $28.67 \pm 2.84$ |
| FLARE-4HOP | $34.87 \pm 3.14$ |

## 5 CONCLUSION

In this study, we exploit physical principles of fluids to develop the FLARE method. FLARE performs graph rewiring with 2-hop connections based on input flow directions. It is a local and directional rewiring method, which is significantly different from the non-directional and long-distance rewiring technique, PIORF, which does not abide the physical principles. Extensive experiments conducted on three datasets and multiple architectures confirmed that FLARE outperforms PIORF and physical principles are critical to the performance.

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

# A  APPENDIX

## A.1  EXPERIMENTAL CONFIGURATIONS

### A.1.1  MODEL ARCHITECTURES

We conduct experiments with three baseline models:

- **MGN:** We make PyTorch reimplementation of the original TensorFlow release, preserving all architectural details including 15 message passing blocks and the original training protocol.
- **BSMS-GNN:** Following the authors' latest release, we configure the model with dataset-specific hierarchical levels—7 levels for *CylinderFlow* and 9 levels for *Airfoil*, as validated in PIORF work.
- **Transolver++:** We adopt the default configuration with additional message passing blocks to better handle mesh-based representations. For incompressible flow datasets (*CylinderFlow* and *Tandem-Airfoil-Cruise*), we add 2 message passing blocks, while for the compressible *Airfoil* dataset, we add 1 message passing block to the base architecture.

### A.1.2  TRAINING DETAILS

Table 5 summarizes our training configurations. We use unit batch size for *CylinderFlow* and *Tandem-Airfoil-Cruise* due to various mesh topologies across simulations, and for MGN to maintain consistency with the original implementation. For *Airfoil*, where mesh structures are uniform, we scale batch size to maximize GPU utilization. In addition, throughout our base FLARE experiments, we employ a zero threshold ($T = 0$) for flow-aligned rewiring, retaining all candidate edges with positive velocity projections.

### A.1.3  *Tandem-Airfoil-Cruise* PREPROCESSING

For *Tandem-Airfoil-Cruise*, we employ geometry-aware encodings tailored to each architecture: signed distance functions (SDF) for MGN and extended directional integrated distance (DID) (Jessica, 2025) on the dual-body configuration for Transolver++, enabling effective representation of complex tandem airfoil interactions.

Table 5: Experimental configurations for models across datasets

| Dataset | Batch size | | | Noise scale | FLARE $T$ |
|---|---|---|---|---|---|
| | MGN | BSMS-GNN | Transolver++ | | |
| *CylinderFlow* | 1 | 1 | 1 | velocity: 2e-2 | 0 |
| *Airfoil* | 1 | 32 | 24 | velocity: 1e1, density: 1e-2 | 0 |
| *Tandem-Airfoil-Cruise* | 1 | / | 1 | no noise | 0 |

## A.2  ABLATION STUDIES OF FLARE DESIGN CHOICES

In this section, we analyze how FLARE behaves for positive thresholds $T > 0$ on the *CylinderFlow* dataset. We focus on three aspects: (i) how sensitive performance is to the choice of $T$ (threshold sensitivity), (ii) whether 3-hop edges helps compared to using only 2-hop edges (locality), and (iii) whether the flow-alignment rule itself matters beyond simply adding more unidirectional edges (directionality and flow alignment). These questions are addressed by the *Ablation Threshold*, *Ablation Multi-hop*, and *Ablation Unidirectional* variants defined below.

**Flow-alignment score for** $T > 0$**.** When $T > 0$, the flow-alignment score is also used to *rank* and threshold candidates, which more explicitly accounts for both velocity magnitude and the spatial scale of the displacement. For the main ablation experiments reported in Table 6, we therefore apply the distance-normalized score

$$norm\_s_{ij} = \mathbf{v}_i^\top \frac{\mathbf{d}_{ij}}{\|\mathbf{d}_{ij}\|^{3/2}}, \tag{7}$$

to each candidate pair $(i, j)$, where $\mathbf{d}_{ij} = \mathbf{x}_j - \mathbf{x}_i$, and use the same $norm\_s_{ij}$ for both 2-hop and 3-hop candidates (when 3-hop edges are included in the multi-hop ablation). Note that the denominator is strictly positive, so the sign of $norm\_s_{ij}$ is unchanged compared to the flow-alignment score $s_{ij}$ used in the main FLARE configuration. Consequently, with $T = 0$ this normalized score selects exactly the same 2-hop connections as FLARE; the normalization only affects ranking and thresholding when $T > 0$ and when 3-hop candidates are additionally considered.

**Ablation design.**    All ablations are run on *CylinderFlow* with the MGN backbone. We consider thresholds $T \in \{0.2, 0.5, 0.8, 1.0\}$ and design one variant for each of the three aspects above:

- *Ablation Threshold.* To focus on the effect of $T$ itself, we keep the 2-hop candidates and the score $norm\_s_{ij}$ fixed, and only vary the threshold. For a given $T$, we add exactly those 2-hop edges with $norm\_s_{ij} > T$.
- *Ablation Multi-hop.* To investigate whether longer-range edges can substitute for 2-hop locality, we start from the Threshold variant for a given $T$ (fewer 2-hop edges) and then add top-scoring 3-hop edges, ranked by the same score $norm\_s_{ij}$, until the total number of added edges matches FLARE's edge count. This keeps the edge budget fixed while trading some 2-hop edges for 3-hop edges.
- *Ablation Unidirectional.* To test whether FLARE gains from its flow-aligned selection, or merely from having more unidirectional edges, we construct this variant separately for each $T$. For a given $T$, we start from all 2-hop candidates, apply FLARE's alignment rule at that threshold, and exclude the 2-hop connections that the corresponding Threshold variant would select. From the remaining 2-hop candidates, we then select unidirectional edges without enforcing the flow-alignment rule; if this still yields fewer edges than FLARE, we additionally include 3-hop edges until the total number of added edges matches FLARE's edge count.

Table 6: Full-rollout RMSE on *CylinderFlow* for FLARE and ablation variants (mean $\pm$ SE). Values are scaled by $\times 10^3$.

| Method ($norm\_s_{ij}$) | $T = 0$ | $T = 0.2$ | $T = 0.5$ | $T = 0.8$ | $T = 1.0$ |
|---|---|---|---|---|---|
| **FLARE (ours)** | **23.38 $\pm$ 2.50** | - | - | - | - |
| Ablation Threshold | - | 28.94 $\pm$ 3.36 | 31.55 $\pm$ 2.98 | 29.91 $\pm$ 2.61 | 30.56 $\pm$ 3.05 |
| Ablation Multi-hop | - | 32.88 $\pm$ 3.06 | 29.06 $\pm$ 2.86 | 39.84 $\pm$ 3.83 | 25.40 $\pm$ 1.96 |
| Ablation Unidirectional | - | 34.68 $\pm$ 3.25 | 38.80 $\pm$ 3.14 | 36.23 $\pm$ 3.50 | 37.80 $\pm$ 3.50 |

FLARE with $T = 0$ attains the lowest error among all configurations, and all variants with $T > 0$ remain noticeably above this level, indicating that the default $T = 0$ configuration is effective while remaining the simplicity to apply. The Threshold ablation shows that positive $T$ still clearly improves over the baseline graph (see Table 2), but does not surpass $T = 0$, suggesting that retaining all flow-aligned 2-hop edges is preferable in this setting. The Multi-hop ablation indicates that introducing additional 3-hop edges, while keeping the edge budget fixed, does not provide a clear advantage over using 2-hop edges alone, which supports 2-hop locality as a reasonable design choice here. Finally, the Unidirectional ablation, which uses a comparable edge budget but does not enforce flow alignment, is markedly worse than FLARE across all $T$, indicating that the gains come from selectively adding flow-aligned edges rather than merely from increasing the number of unidirectional shortcuts.

**Ablation experiments with $s_{ij}$.**    The ablations above use the distance-normalized flow-alignment score $norm\_s_{ij}$ for $T > 0$, which down-weights longer displacements and therefore prefers shorter, more local connections. In addition to these ablations, we also test the three variants (Threshold, Multi-hop, Unidirectional) with the original, unnormalized score $s_{ij}$ to compare directly against the modified flow-alignment rule for $T > 0$.

For each threshold $T \in \{0.2, 0.5, 0.8, 1.0\}$ used with $norm\_s_{ij}$ in Table 6, we choose a corresponding threshold $\hat{T}(T)$ for $s_{ij}$, such that on average over the training set, the rule $s_{ij} > \hat{T}(T)$ retains a similar proportion of 2-hop candidates as $norm\_s_{ij} > T$. The resulting mapping is summarized in Table 7, and the associated rollout errors for the three ablation variants are reported in Table 8. Taken together, the experiments with $s_{ij}$ further confirm the conclusions drawn from

the $norm\_s_{ij}$ ablations, while validating that the distance-normalized score $norm\_s_{ij}$ is the better choice. It consistently achieves lower rollout error under comparable rewiring and works as intended by prioritizing edges with strong, well-aligned velocities while penalizing large $\|\mathbf{d}_{ij}\|$, which helps discourage long jumps that tend to hurt performance when higher-hop candidates are present. Given these benefits, we adopt $norm\_s_{ij}$ as our flow-alignment score for $T > 0$.

Table 7: Mapping from thresholds $T$ used with the normalized score $norm\_s_{ij}$ to the corresponding thresholds $\hat{T}(T)$ for the original score $s_{ij}$ on *CylinderFlow*.

| $T$ | $\hat{T}(T)$ |
|-----|-----|
| 0.2 | 0.0014 |
| 0.5 | 0.0036 |
| 0.8 | 0.0058 |
| 1.0 | 0.0079 |

Table 8: Full-rollout RMSE on *CylinderFlow* for ablation variants using $s_{ij}$ (mean $\pm$ SE). Values are scaled by $\times 10^3$.

| Method ($s_{ij}$) | $\hat{T}(0.2)$ | $\hat{T}(0.5)$ | $\hat{T}(0.8)$ | $\hat{T}(1.0)$ |
|-----|-----|-----|-----|-----|
| Ablation Threshold | $32.14 \pm 3.16$ | $34.40 \pm 2.83$ | $36.66 \pm 1.94$ | $36.52 \pm 3.32$ |
| Ablation Multi-hop | $36.53 \pm 3.07$ | $33.83 \pm 3.11$ | $28.02 \pm 2.52$ | $40.54 \pm 3.95$ |
| Ablation Unidirectional | $38.19 \pm 2.99$ | $52.09 \pm 3.09$ | $39.13 \pm 3.11$ | $36.74 \pm 2.94$ |

### A.3 RUNTIME AND MEMORY ANALYSIS

In this section, we report per-step training and inference time, as well as peak GPU memory usage, for *CylinderFlow* on the MGN backbone with different rewiring schemes. All timings are measured on the same hardware and implementation as in the main experiments. For each method, we discard an initial warm-up phase and then average over 50 iterations. The offline computation of Ollivier–Ricci curvature (required by PIORF) is not included in the timings below.

Table 9: Per-step training and inference time (mean $\pm$ SD) on *CylinderFlow* (MGN backbone).

| Model | Training (ms) | Inference (ms) | Train vs MGN | Infer vs MGN |
|-----|-----|-----|-----|-----|
| MGN (Baseline) | $52.47 \pm 8.76$ | $30.44 \pm 7.74$ | $1.000\times$ | $1.000\times$ |
| PIORF (3% ORC) | $58.65 \pm 9.16$ | $36.38 \pm 9.61$ | $1.118\times$ | $1.195\times$ |
| 2-HOP-ALL | $97.84 \pm 7.14$ | $33.56 \pm 8.51$ | $1.865\times$ | $1.103\times$ |
| 2-HOP-RANDOM | $58.06 \pm 9.21$ | $32.10 \pm 7.90$ | $1.107\times$ | $1.054\times$ |
| FLARE (ours) | $67.79 \pm 8.33$ | $39.54 \pm 9.51$ | $1.292\times$ | $1.299\times$ |

### A.4 EXTENSION: SEPARATE REWIRING FOR DENSITY ON COMPRESSIBLE *Airfoil*

From the results in Table 2, we observe that on the compressible *Airfoil* dataset, FLARE yields notable improvements for velocity prediction, while the relative gain on density is smaller. In this section, we propose a simple extension in which we introduce a separate 2-hop based rewiring for density and use it in parallel with the FLARE rewiring for velocity.

Start from graph $G = (V, E)$, 2-hop candidate set $C_2$, and $A_{\text{align}}$ as selected connections by FLARE following the flow-alignment rule $s_{ij} > T$ as defined in Section 3. For velocity prediction, we continue to use the rewired edge set $E \cup A_{\text{align}}$. For density prediction, we introduce an additional selected 2-hop connections $A_\rho$ built on top of the same base edges $E$, such that density uses the edge set $E \cup A_\rho$.

To construct $A_\rho$, we consider all node pairs $(i, j) \in C_2$ and define

$$\Delta\rho_{ij} = |\rho_i - \rho_j|, \tag{8}$$

where $\rho_i$ and $\rho_j$ are the density values at nodes $i$ and $j$. We rank all 2-hop pairs by $\Delta\rho_{ij}$ and select the top 10% pairs. For each selected pair $(i, j)$, we make a bidirectional connection by adding both directed edges $(i, j)$ and $(j, i)$ to $A_\rho$ so that density information can propagate in both directions between neighbouring regions with strong local contrast.

Table 10: Edge counts and peak GPU memory usage on *CylinderFlow* (MGN backbone).

| Model | Params | Base Edges | Added Edges | Total Edges | Peak Mem. (GB) | vs MGN |
|---|---|---|---|---|---|---|
| MGN (Baseline) | 2,332,930 | 10,488 | 0 | 10,488 | 0.67 | 1.00× |
| PIORF (3% ORC) | 2,332,933 | 10,488 | 109 | 10,597 | 0.68 | 1.02× |
| 2-HOP-ALL | 2,332,930 | 10,488 | 20,378 | 30,866 | 1.84 | 2.76× |
| 2-HOP-RANDOM | 2,332,930 | 10,488 | 10,928 | 21,416 | 1.27 | 1.90× |
| FLARE (ours) | 2,332,930 | 10,488 | 10,928 | 21,416 | 1.27 | 1.90× |

In the model with separate rewiring, each message passing layer uses two edge sets on the same node set $V$: $E \cup A_{\text{align}}$ for velocity updates and $E \cup A_\rho$ for density updates. At each layer, we perform separate message passing on these two edge sets using the current node features, and then combine the two updated feature vectors at each node with a small MLP to obtain the input for the next layer. We apply this extension to MGN, BSMS-GNN, and Transolver++ on the *Airfoil* dataset.

Table 11 reports the resulting full-rollout RMSE for velocity and density. The same numbers are also included in the main *Airfoil* results (Table 2) as the "FLARE + 10% Density 2HOP" rows, where this extension achieves the lowest velocity and density RMSE among all compared methods for each of the three backbones on this dataset, suggesting that FLARE can be extended with improvements by introducing separate rewiring for additional fields such as density.

Table 11: FLARE extension with separate rewiring for density on *Airfoil*. Values are full-rollout RMSE (mean $\pm$ SE); velocity is reported in original scale and density is scaled by $\times 10^3$.

| Method | Velocity RMSE | Density RMSE ($\times 10^3$) |
|---|---|---|
| MGN (FLARE + 10% Density 2HOP) | $31.93 \pm 2.57$ | $85.66 \pm 6.12$ |
| BSMS-GNN (FLARE + 10% Density 2HOP) | $39.46 \pm 2.90$ | $95.58 \pm 7.05$ |
| Transolver++ (FLARE + 10% Density 2HOP) | $34.26 \pm 2.32$ | $64.76 \pm 4.71$ |

## A.5 FLOW DYNAMICS ANALYSIS ON *CylinderFlow*

In this section, we investigate how FLARE's improvements relate to local flow dynamics on the *CylinderFlow* dataset. For each node, we compute the average angular change in velocity direction between consecutive timesteps over 100 rollout trajectories in the test set. Nodes are then sorted by this average angular change and divided into five equal-sized quintiles, from the least to the most dynamically varying. The resulting ranges are summarized in Table 12.

Table 12: Quintile ranges of average velocity-direction change on *CylinderFlow*.

| Quintile | $\theta$ Range (avg) |
|---|---|
| Q1 | $0.00° - 0.03°$ |
| Q2 | $0.03° - 0.04°$ |
| Q3 | $0.04° - 0.11°$ |
| Q4 | $0.11° - 0.63°$ |
| Q5 | $0.63° - 90.00°$ |

For each quintile, we evaluate the per-node rollout RMSE of different rewiring methods based on the MGN backbone. Table 13 reports the errors and relative improvements over the MGN baseline.

The quintile analysis shows that FLARE achieves the lowest error and the largest improvement over the baseline across all dynamics levels. In particular, the highest improvement (45.4%) occurs in the most dynamic quintile (Q5). This supports our observation that FLARE provides particularly strong benefits in regions with evolving flow patterns, such as the unsteady wake structures in *Cylinder-Flow*.

Table 13: Per-node RMSE and relative improvement (%) by flow-dynamics quintiles on *Cylinder-Flow*.

| | Q1 | | Q2 | | Q3 | | Q4 | | Q5 | |
|---|---|---|---|---|---|---|---|---|---|---|
| Method | RMSE | Impr. | RMSE | Impr. | RMSE | Impr. | RMSE | Impr. | RMSE | Impr. |
| Baseline (MGN) | 12.61 | - | 9.95 | - | 13.16 | - | 47.59 | - | 51.99 | - |
| PIORF | 11.61 | 8.0% | 9.74 | 2.1% | 10.49 | 20.3% | 36.25 | 23.8% | 43.05 | 17.2% |
| 2-HOP-ALL | 10.68 | 15.3% | 8.16 | 18.0% | 10.70 | 18.7% | 33.88 | 28.8% | 34.16 | 34.3% |
| 2-HOP-RANDOM | 10.79 | 14.5% | 8.19 | 17.7% | 11.25 | 14.6% | 40.91 | 14.0% | 40.54 | 22.0% |
| FLARE (ours) | **7.34** | **41.8%** | **5.87** | **41.0%** | **8.04** | **38.9%** | **30.88** | **35.1%** | **28.40** | **45.4%** |

## A.6 ROLLOUT PREDICTION ERROR VISUALIZATION

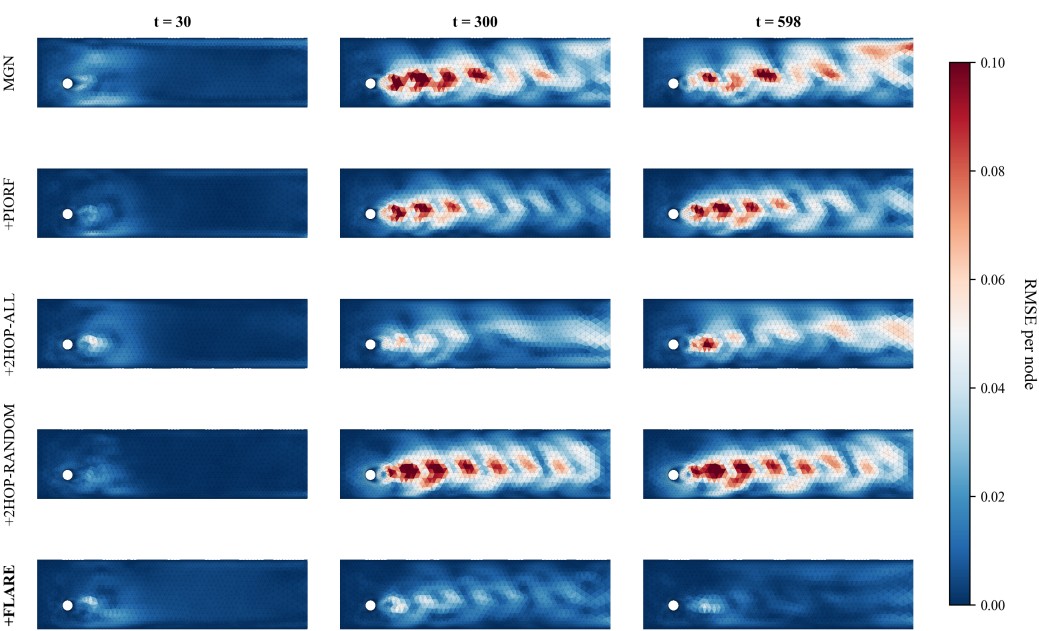

Figure 5: Rollout prediction errors at timesteps t = {30, 300, 598} for different graph rewiring methods on *CylinderFlow* dataset. Color intensity indicates error magnitude.

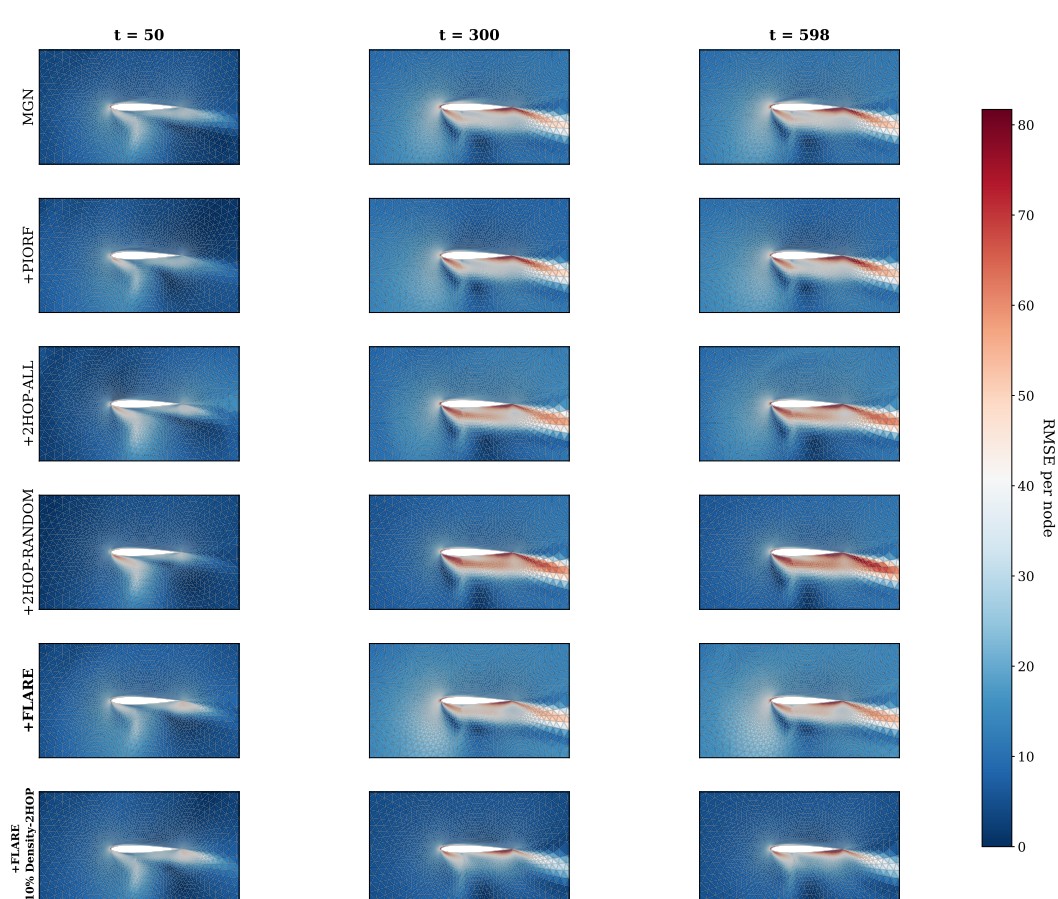

Figure 6: Rollout prediction errors (velocity) at timesteps t = {30, 300, 598} for different graph rewiring methods on *Airfoil* dataset. Color intensity indicates error magnitude.

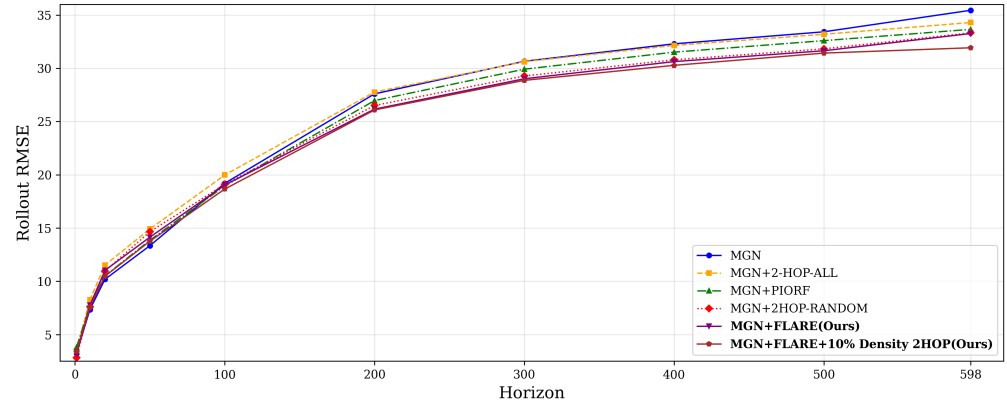

Figure 7: Rollout prediction error (velocity) on *Airfoil* test set.

