# OpenReview forum: "Graph Rewiring based on Flow Alignment for Improving Fluid Simulation"
_ICLR.cc/2026/Conference — Submitted to ICLR 2026_

### Official Review · Reviewer_5y92 · 2025-10-16

**Soundness:** 2
**Presentation:** 2
**Contribution:** 2
**Rating:** 4
**Confidence:** 4

**Summary:**

The paper proposes FLARE, a simple graph rewiring strategy that is 2-hop, flow-aligned, directional, and updated each time step. At every step, it adds new along-flow edges within the local 2-hop neighborhood, so message passing follows physical advection and reduces over-squashing. FLARE is easy to plug into different backbones and shows lower long-horizon error, outperforming static or long-range baselines such as 2-HOP-ALL and PIORF on cylinder and airfoil benchmarks.

**Strengths:**

A lightweight, plug-and-play rule that rewires 2-hop, flow-aligned, directional edges dynamically well-matched to advective transport and slows long-horizon error growth.

**Weaknesses:**

Relies on strict unidirectionality and a fixed 2-hop scope (no CFL-based adaptivity, vortex/backflow handling, or conservation metrics), and lacks fair comparisons against directional/dynamic PIORF under matched edge budgets.

**Questions:**

1. In practical CFD, during one time step, roughly how many graph hops does advection cover? If the mesh/flow is anisotropic (Δx differs by direction), should you keep one hop count everywhere?
2. In regions with backflow or vortices, does the strict unidirectional downwind assumption break down?
3. From a physical standpoint, can PIORF’s rewiring be interpreted as an approximation to a non-local operator?
4. Can PIORF be implemented in directional and/or dynamic variants, and are there comparative results against related methods?
5. Do local unidirectional edge additions risk violating conservation constraints (e.g., mass, divergence-free, energy)?

---

> ### Author Response · Authors · 2025-11-24
> **Response to Reviewer 5y92 (1/2)**
>
> Dear Reviewer 5y92,
>
> Thank you very much for your review and your recognition that FLARE is a lightweight, plug-and-play rewiring rule whose dynamic 2-hop, flow-aligned directional edges are well matched to advective transport, help slow long-horizon error growth, and integrate easily into different backbones.
>
> We will carefully address each of your questions and comments below.
>
> ---
>
> **Q1. In practical CFD, during one time step, roughly how many graph hops does advection cover? If the mesh/flow is anisotropic ($\\Delta x$ differs by direction), should you keep one hop count everywhere?**
>
> **"In practical CFD, during one time step, roughly how many graph hops does advection cover?"**
>
> For CFD simulation, the time step is chosen to satisfy a standard Courant--Friedrichs--Lewy (CFL) condition of the form
>
> $$C_o = \\frac{|u| \\,\\Delta t}{\\Delta x_{\\text{local}}} < 1,$$
>
> where $\\Delta x_{\\text{local}}$ is the local cell size in the flow direction. This implies that, in physical space, a fluid parcel typically travels less than one cell spacing along the most restrictive direction during a single time step. So, to answer technically, even one hop will always be greater than the physical distance $|u| \\,\\Delta t$.
>
> However, it does not mean that CFD or physics-informed AI networks can achieve accurate simulation through one iteration or one layer of GNN computation.
>
> **"If the mesh/flow is anisotropic ($\\Delta x$ differs by direction), should you keep one hop count everywhere?"**
>
> For CFD, keeping one hop count everywhere should be no problem. As the mesh is anisotropic, the local cell size is changed/refined according to flow condition already, so the CFL condition is satisfied.
>
> For physics-informed AI, we can keep using one hop count everywhere, like the baseline methods. However, in this work, we have demonstrated that FLARE outperforms the baselines significantly through physics-informed rewiring.
>
> ---
>
> **Q2. In regions with backflow or vortices, does the strict unidirectional downwind assumption break down?**
>
> Thank you for your question about directionality. We would like to clarify that FLARE does not impose a strict unidirectional assumption. Rather, the directionality of connections emerges naturally from the flow field and our flow alignment-based selection criterion.
>
> Our method evaluates each potential 2-hop edge independently based on whether the sender's velocity aligns with the displacement vector ($s_{ij} > T$). In regions with predominantly unidirectional flow, this naturally produces mostly unidirectional connections. However, in regions with backflow, recirculation, or vortices where flow directions are more complex, bidirectional connections can and do emerge naturally when both $(i,j)$ and $(j,i)$ satisfy the alignment criterion.
>
> To quantify this, we analyzed the directionality of selected edges across 10 trajectories through all timesteps from the Cylinder Flow dataset:
>
> - Average total directed edges selected: 9,643.8
> - Average bidirectional edges: 150.8 × 2
> - Average unidirectional edges: 9,342.1
>
> These statistics show that while the majority of connections are unidirectional, bidirectional connections do occur naturally where the flow structure supports them. The bidirectional connections likely correspond to regions with backflow, vortices, or recirculation zones where flow alignment permits connections in both directions.
>
> Figure 1c in our manuscript was based on real data, which visualizes bidirectional connections (shown in red), illustrating how FLARE adapts to local flow patterns.
>
> ---
>
> **Q3. From a physical standpoint, can PIORF's rewiring be interpreted as an approximation to a non-local operator?**
>
> PIORF uses curvature and feature-space velocity differences to add bidirectional long-range edges, with the explicit goal of alleviating over-squashing and improving information flow on the mesh. We do observe that PIORF outperforms the baseline, but the added connections are not derived from the Navier–Stokes equations, do not correspond to a known non-local kernel, and are not constrained by physical properties such as locality, conservation, or one-way transport along the flow. Instead, PIORF's bidirectional non-local connections can be viewed as a numerical AI mechanism to pass information between distant nodes more quickly, similar in spirit to other non-local neural architectures and rewiring techniques in the ML literature (e.g., non-local neural networks, transformer-style global attention, and curvature-based oversquashing mitigation in GNNs). These techniques are effective at speeding up information propagation on the graph, but their benefits come from generic message-passing improvements rather than from approximating a physically grounded non-local fluid operator.
>
> ---

---

> ### Author Response · Authors · 2025-11-24
> **Response to Reviewer 5y92 (2/2)**
>
> **Q4. Can PIORF be implemented in directional and/or dynamic variants, and are there comparative results against related methods?**
>
> Thank you for this question.
>
> We would like to clarify that PIORF is already a dynamic method: it does dynamic rewiring at every timestep, using offline Ollivier-Ricci-Curvature (ORC)-based seeds plus the current velocity field projected to 1D.
>
> While the PIORF method was proposed to be bidirectional, we did experiment *PIORF Aligned*, which maintains PIORF's 3% ratio for edge addition (claimed as optimal for Cylinder Flow in the original work) but restricts added edges to only those aligned with the flow direction. The results on Cylinder Flow are:
>
> | Method | Rollout RMSE |
> |:------------------|:----------------------:|
> | PIORF (original) | 33.59 ± 3.70 |
> | PIORF Aligned (directional variant) | 32.90 ± 3.46 |
>
> The directional variant shows a slight difference but still underperforms FLARE significantly, which combines flow alignment with 2-hop locality.
>
> ---
>
> **Q5. Do local unidirectional edge additions risk violating conservation constraints (e.g., mass, divergence-free, energy)?**
>
> Thank you for this thought-provoking question.
>
> We respectfully disagree that the violation of conservation constraints is a concern as our approach, despite learning with prior physics knowledge, is still mainly a data-driven method. In other words, with or without the local unidirectional edge additions, our approach does not consider any conservation constraints anyway. However, improving the accuracy of our model predictions relative to the ground truths, which were generated from CFD solver that preserves conservation laws, would suggest that we are closer to satisfying the constraints. Furthermore, recognizing that the conservation constraints are essentially given by the governing transport equations (e.g., divergence-free constraint is the conservation of mass equation in incompressible flow), we believe that there are ways where we can construct training loss functions to guide the model to obey conservation constraints, a feature that has been attained in physics-informed neural network (PINN) approach.

---

> ### Author Response · Authors · 2025-11-28
> **Gentle Reminder**
>
> Dear Reviewer 5y92,
>
> We hope you are doing well. We are just reaching out with a gentle follow‑up regarding our earlier response and to inquire if you have been able to look it over.
>
> Whenever you have a moment to share your comments or confirmation, it would be much appreciated.
>
> Thank you once more for your time and valuable review.
>
> Kind regards,
> Authors of the Submission 15260

---

### Official Review · Reviewer_ebA5 · 2025-10-24

**Soundness:** 2
**Presentation:** 3
**Contribution:** 2
**Rating:** 4
**Confidence:** 3

**Summary:**

The authors explore graph rewiring for GNN-based surrogates for CFD modeling. They note that previous rewiring schemes and PIORF' long-range connections ignore fluid dynamics and can violate physical principles of fluid dynamics. Authors propose their FLARE method (Flow alighment rewiring) that adds selected 2-hop neighbours and uses dot product between sender's velocity and the displacement vector to decide if a directed edge should be added. This methodology aligns flow with the velocity and graph is rewired at each time step. Experiments on three datasets show that simply adding all 2-hop neighbors already rivals PIORF, and FLARE offers gains. Ablations show that adding edges opposite to the flow decreases performance and 3/4-hop rewiring also degrades results.

**Strengths:**

The idea is novel, the paper is well-written, good presentation, also:
1. The physics-informed rewiring heuristic uses simple fluid-dynamics principles and often gives a gain in performance.
2. The method is easy to implement.
3. The authors made ablation study.

**Weaknesses:**

1. Gains of FLARE are dataset-dependent sometimes. But the paper draws broad conclusions. For example, "By comparing FLARE with 2-HOP-ALL and PIORF,we can conclude the effectiveness of physics guided design. The directional and local connections determined by flow directions are essential to achieve these performance gains."
2. Authors have not analyzed threshold parameter sensitivity. It is simply fixed to T=0 in main experiments as I understand. Therefore, ablation study is limited.
3. No runtime/memory cost analysis was performed.

**Questions:**

1. Make more solid explanation of superiority of FLARE. See W1. Try not to use vague proves. Authors explain dataset dependency using this sentence "It is worth noting that FLARE gains more significant improvements on CylinderFlow than Airfoil likely because the former has more dynamics and challenging  flow conditions, giving more room for FLARE to improve."  Could the authors provide a more detailed analysis of why FLARE benefits more from dynamic flows, and how the method might be adapted to perform better on simpler cases such as the Airfoil density prediction?
2. Analyze flow-alighment threshold parameter sensitivity.
3. Conduct runtime/memory cost analysis.
4. Can you show rollout error vs. time for Airfoil and Tandem-Airfoil, not only CylinderFlow?

---

> ### Author Response · Authors · 2025-11-24
> **Response to Reviewer ebA5 (1/3)**
>
> Dear Reviewer ebA5,
>
> We sincerely appreciate your careful assessment and your recognition of the novelty of our idea, the clarity of the presentation, and the fact that our physics-informed 2-hop rewiring is both grounded in simple fluid-dynamics principles, easy to implement, and supported by ablation studies.
>
> We address your questions and comments point by point below.
>
> ---
> **W1 & Q1**
>
> **W1. Gains of FLARE are dataset-dependent sometimes. But the paper draws broad conclusions. For example, "By comparing FLARE with 2-HOP-ALL and PIORF, we can conclude the effectiveness of physics guided design. The directional and local connections determined by flow directions are essential to achieve these performance gains."**
>
> **Q1. Make more solid explanation of superiority of FLARE. See W1. Try not to use vague proves. Authors explain dataset dependency using this sentence "It is worth noting that FLARE gains more significant improvements on CylinderFlow than Airfoil likely because the former has more dynamics and challenging flow conditions, giving more room for FLARE to improve." Could the authors provide a more detailed analysis of why FLARE benefits more from dynamic flows, and how the method might be adapted to perform better on simpler cases such as the Airfoil density prediction?**
>
> We thank the reviewer for this important critique of our claims. We acknowledge that our original statement was overly broad. We have revised the broad conclusion in the manuscript. The new statement reads: "The experimental results on the three datasets indicate that FLARE outperforms 2-HOP-ALL and PIORF. These results demonstrate the effectiveness of our design combining flow alignment, directionality, and 2-hop locality. FLARE achieves consistent improvements across different flow dynamics regimes in the three datasets, with particularly strong benefits in regions with evolving flow patterns."
>
> To provide a more rigorous analysis of how FLARE's benefits relate to flow dynamics, we conducted a detailed study categorizing nodes based on their flow dynamics. Specifically, we computed the average angular change in velocity direction between consecutive timesteps for each node across 100 trajectories in the Cylinder Flow test set, then divided all nodes into five equal quintiles based on their dynamics levels:
>
> **Table: Quintile ranges of average angular change in velocity direction ($\\theta$)**
>
> | Quintile | $\\theta$ Range (avg) |
> |:-----------:|:-----------------------------:|
> | Q1 | $0.00°$ -- $0.03°$ |
> | Q2 | $0.03°$ -- $0.04°$ |
> | Q3 | $0.04°$ -- $0.11°$ |
> | Q4 | $0.11°$ -- $0.63°$ |
> | Q5 | $0.63°$ -- $90.00°$ |
>
> We then evaluated per-node RMSE for each quintile:
>
> | Method | Q1 | Q2 | Q3 | Q4 | Q5 |
> |:-------------|:---------------:|:---------------:|:---------------:|:---------------:|:---------------:|
> | Baseline (MGN) | 12.610 | 9.948 | 13.162 | 47.594 | 51.989 |
> | PIORF | 11.605 | 9.736 | 10.491 | 36.246 | 43.046 |
> | 2-HOP-ALL | 10.682 | 8.157 | 10.704 | 33.883 | 34.164 |
> | 2-HOP-RANDOM | 10.787 | 8.189 | 11.247 | 40.909 | 40.543 |
> | FLARE (ours) | **7.340** | **5.870** | **8.041** | **30.879** | **28.396** |
>
> | Method | Q1 | Q2 | Q3 | Q4 | Q5 |
> |:-------------|:---------------:|:---------------:|:---------------:|:---------------:|:---------------:|
> | PIORF | 8.0% | 2.1% | 20.3% | 23.8% | 17.2% |
> | 2-HOP-ALL | 15.3% | 18.0% | 18.7% | 28.8% | 34.3% |
> | 2-HOP-RANDOM | 14.5% | 17.7% | 14.6% | 14.0% | 22.0% |
> | FLARE (ours) | **41.8%** | **41.0%** | **38.9%** | **35.1%** | **45.4%** |
>
> The quintile analysis shows that FLARE achieves relatively larger improvements in more dynamic regions, with the highest improvement (45.4%) occurring in the most dynamic quintile (Q5). Note that FLARE offers much more improvements in Q4 and Q5 in terms of RMSE since the baseline has much higher errors in these two quintiles. Cylinder Flow exhibits more dynamic patterns (vortex shedding, unsteady wake structures) compared to Airfoil's relatively steadier flow, which is consistent with the larger overall gains observed on Cylinder Flow.
>
> We have included this analysis in Appendix A.5 of our revised manuscript.

---

> ### Author Response · Authors · 2025-11-24
> **Response to Reviewer ebA5 (2/3)**
>
> **Q2. Analyze flow-alignment threshold parameter sensitivity.**
>
> Thank you for the question regarding the threshold ($T$) sensitivity test.
>
> Taking the feedback about velocity magnitude from the reviewer UERp together, we have revised our flow alignment score formulation with distance-dependent normalization: $norm\\_s\_{ij} = \\mathbf{v}\_i^T \\dfrac{\\mathbf{d}\_{ij}}{\\lVert \\mathbf{d}\_{ij} \\rVert \\cdot \\lVert \\mathbf{d}\_{ij} \\rVert^{0.5}}$. Note that when $T=0$ (FLARE), the original score formula and revised one will provide the same 2-hop connections. All ablation studies below use this revised formulation.
>
> To examine threshold parameter sensitivity, we conducted experiments with $T \\in \\{0.2, 0.5, 0.8, 1.0\\}$ on the Cylinder Flow dataset:
>
> **Table: Threshold sensitivity and multi-hop ablations on CylinderFlow**
>
> | Method | $T=0$ | $T=0.2$ | $T=0.5$ | $T=0.8$ | $T=1.0$ |
> |:-----------------|:---------------------:|:----------------------:|:---------------------:|:--------------------:|:--------------------:|
> | **FLARE (ours)** | **23.38 ± 2.50** | - | - | - | - |
> | Ablation Threshold | - | 28.94 ± 3.36 | 31.55 ± 2.98 | 29.91 ± 2.61 | 30.56 ± 3.05 |
> | Ablation Multihop | - | 32.88 ± 3.06 | 29.06 ± 2.86 | 39.84 ± 3.83 | 25.40 ± 1.96 |
>
> **Ablation Threshold:** Varies $T$ to select only 2-hop edges with $norm\\_s\_{ij} > T$, naturally reducing the number of selected edges as $T$ increases. This directly tests threshold sensitivity while keeping locality and directionality constant.
>
> **Ablation Multi-hop:** Uses threshold $T$ for 2-hop selection, then adds top-scoring 3-hop edges to maintain the same total edge count as FLARE. This tests whether fewer 2-hop edges can be compensated by longer-range connections.
>
> The results show that FLARE with $T=0$ achieves the best performance (full rollout RMSE: 23.38) compared to other threshold variants. Performance remains reasonably stable across different $T$ values (for Ablation Threshold), demonstrating robustness to threshold selection. The Multi-hop ablation shows more variability, indicating that compensating with 3-hop edges cannot fully replace the effectiveness of flow-aligned 2-hop edges. FLARE with $T=0$ is also the most straightforward configuration, which is easy to implement.
>
> We have also conducted Ablation Unidirectional experiments (detailed in response to Reviewer PxA5) to isolate the contribution of flow alignment versus mere directionality, further validating our design choices.
>
> We have updated these ablation studies into our revised manuscript (Appendix A.2) with detailed ablation designs and experiment results analysis.

---

> ### Author Response · Authors · 2025-11-24
> **Response to Reviewer ebA5 (3/3)**
>
> **Q3. Conduct runtime/memory cost analysis.**
>
> We appreciate the directive to conduct a runtime and memory cost analysis. Accordingly, we measured per-step training and inference time and peak memory on Cylinder Flow, averaging over 50 iterations on the same device used in the paper.
>
> **Timing details:** For all two hop variants, including FLARE, we compute two-hop candidates and perform edge selection online during training, so the reported times include this cost. In contrast, PIORF requires an offline Ollivier--Ricci curvature (ORC) ranking; computing the ORC ranking for the Cylinder Flow training set takes ~40 minutes, which is not included in the per-step times below (to avoid conflating one-time preprocessing with per-step training).
>
> **Table 1: Training and Inference Time Comparison**
>
> | Model | Training (ms) | Inference (ms) | Train vs MGN | Infer vs MGN |
> |:---------|:-----------------:|:------------------:|:-----------------:|:----------------:|
> | MGN (Baseline) | 52.47 ± 8.76 | 30.44 ± 7.74 | 1.000x | 1.000x |
> | PIORF (3% ORC) | 58.65 ± 9.16 | 36.38 ± 9.61 | 1.118x | 1.195x |
> | 2-HOP-ALL | 97.84 ± 7.14 | 33.56 ± 8.51 | 1.865x | 1.103x |
> | 2-HOP-RANDOM | 58.06 ± 9.21 | 32.10 ± 7.90 | 1.107x | 1.054x |
> | FLARE (ours) | 67.79 ± 8.33 | 39.54 ± 9.51 | 1.292x | 1.299x |
>
> **Table 2: Edge Addition and Memory Usage Comparison**
>
> | Model | Params | Base Edges | Added Edges | Total Edges | Peak Memory (GB) | vs MGN |
> |:----------|:--------------:|:---------------:|:---------------:|:---------------:|:-----------------------:|:-----------:|
> | MGN (Baseline) | 2,332,930 | 10,488 | 0 | 10,488 | 0.67 | 1.00x |
> | PIORF (3% ORC) | 2,332,933 | 10,488 | 109 | 10,597 | 0.68 | 1.02x |
> | 2-HOP-ALL | 2,332,930 | 10,488 | 20,378 | 30,866 | 1.84 | 2.76x |
> | 2-HOP-RANDOM | 2,332,930 | 10,488 | 10,928 | 21,416 | 1.27 | 1.90x |
> | FLARE (ours) | 2,332,930 | 10,488 | 10,928 | 21,416 | 1.27 | 1.90x |
>
> In summary, FLARE introduces a moderate and predictable overhead, and in return yields substantial performance gains in our main results **(up to 27.40% over PIORF and 26.03% over 2-HOP-ALL)**. We therefore view the cost--accuracy trade-off as balanced and worthwhile for practical simulation workloads.
>
> We have added the runtime and memory cost analysis in Appendix A.3 of our revised manuscript.
>
> ---
>
> **Q4. Can you show rollout error vs. time for Airfoil and Tandem-Airfoil, not only CylinderFlow?**
>
> We thank the reviewer for this suggestion. In addition to the rollout error curves already shown for *CylinderFlow*, we have now included rollout error curves and a rollout comparison visualization for *Airfoil* in Appendix A.6 of the revised manuscript, with visualization plotted in the same format for direct comparison. The visualization figure illustrates how the error evolves over time across the different rewiring schemes on *Airfoil*. We need to clarify that *Tandem-Airfoil-Cruise* dataset consists of steady state flow simulations, which does not support rollout over timesteps.

---

> > ### Comment · Reviewer_ebA5 · 2025-11-27
> >
> > Thank you for your thorough consideration of my comments. I am satisfied with the responses.

---

### Official Review · Reviewer_PxA5 · 2025-11-01

**Soundness:** 1
**Presentation:** 3
**Contribution:** 2
**Rating:** 4
**Confidence:** 5

**Summary:**

The paper proposes FLARE, a physics-informed GNN rewiring method for fluid simulation that selectively adds directional 2-hop edges based on instantaneous flow alignment. The authors show good performance over prior methods like PIORF and generic 2-hop rewiring.

**Strengths:**

1. The paper demonstrates consistent and substantial performance improvements over PIORF and structural baselines across three diverse datasets (unsteady, steady, compressible/incompressible) and 3 architectures.
2. The authors present a more straightforward solution than existing physics-based rewiring methods.
3. The introduction clearly articulates what problems of existing methods the paper aims to solve.

**Weaknesses:**

1. The proposed FLARE and its evaluation approach are insufficient to support the authors' claimed motivation of "rigorous adherence to physical principles."

2. The use of a zero threshold ($T=0$) raises questions about whether structural directionality (directional 2-hop) is the primary driver of performance rather than the alignment principle itself. Thus, the paper weakens the rigor of the "physics-informed" claim.

3. There is insufficient fluid dynamics or GNN-theoretical (e.g., curvature, over-squashing) justification for determining 2-hop as the optimal locality. The paper presents only experimental results.

4. The computational cost of dynamic graph reconstruction at every time step during rollout (inference time overhead) is not analyzed.

**Questions:**

Q1. What is the rationale for using $T=0$ in the base FLARE experiments, and do you believe this sufficiently tests the physical alignment principle? What happens in the case of positive thresholds where $T>0$?

Q2. More evidence is needed to support the claim that PIORF "does not align with physical principles" (line 237, footnote 2). The differences between FLARE and PIORF need to be discussed more clearly, and the velocity gradient (strain rate) aspect of PIORF should be addressed. Can you explain more precisely which aspects of PIORF are less physically straightforward?

Q3. Why does FLARE underperform PIORF on Airfoil density with BSMS-GNN?

Q4. Please provide a detailed computational cost comparison between baseline, PIORF, 2-HOP-ALL, and FLARE for each dataset. How much overhead does dynamic rewiring add?

Q5. Your comparison confounds multiple factors (locality, directionality, selection criterion). Can you provide ablations that isolate each factor? Specifically, factors such as whether all connections other than inverse connections are directional, or the proportion of bidirectional connections?

Q6. What are the detailed settings for 3-hop and 4-hop in the ablation study?

---

> ### Author Response · Authors · 2025-11-24
> **Response to Reviewer PxA5 (1/5)**
>
> Dear Reviewer PxA5,
>
> We appreciate your thoughtful review and are encouraged by your observation that FLARE achieves consistent and substantial performance improvements over PIORF and structural baselines across three diverse datasets and three architectures, that our physics-informed rewiring remains simpler than existing methods, and that the introduction clearly explains the issues in prior work that we aim to address.
>
> We address your questions and comments point by point below.
>
> ---
>
> **Q1 & Q5 & W3**
>
> **Q1. What is the rationale for using $T = 0$ in the base FLARE experiments, and do you believe this sufficiently tests the physical alignment principle? What happens in the case of positive thresholds where $T > 0$?**
>
> **Q5. Your comparison confounds multiple factors (locality, directionality, selection criterion). Can you provide ablations that isolate each factor? Specifically, factors such as whether all connections other than inverse connections are directional, or the proportion of bidirectional connections?**
>
> **W3. There is insufficient fluid dynamics or GNN-theoretical (e.g., curvature, over-squashing) justification for determining 2-hop as the optimal locality. The paper presents only experimental results.**
>
> Thank you for your questions about our design choices and experimental validation. We address these related concerns together as they all pertain to validating our core design principles.

---

> ### Author Response · Authors · 2025-11-24
> **Response to Reviewer PxA5 (2/5)**
>
> **Q1 & Q5 & W3 continued**
>
> Taking the feedback about velocity magnitude from the reviewer UERp together, we have modified our flow alignment score formula to better handle spatial scale variations. The updated score incorporates distance-dependent normalization: $norm\\_s\_{ij} = \\mathbf{v}\_i^T \\dfrac{\\mathbf{d}\_{ij}}{\\lVert \\mathbf{d}\_{ij} \\rVert \\cdot \\lVert \\mathbf{d}\_{ij} \\rVert^{0.5}}$ (for ablation studies below). All ablation studies reported below use this modified score formulation. Note that when $T = 0$, the original score formula and modified one will provide the same 2-hop connections.
>
> ---
>
> **Q1: Rationale for $T = 0$ and Threshold Sensitivity**
>
> We chose $T = 0$ for our main experiments to test the core principle of flow-alignment-rewiring in its simplest but representative form. By selecting all 2-hop edges where the sender's velocity aligns with the displacement vector (corresponding to $s_{ij} > 0$), we establish whether incorporating flow-aligned edges improves prediction performance compared to baseline graph structures, without introducing additional hyperparameter complexity.
>
> We acknowledge the reviewer's concern that a single threshold choice may not fully validate the physical alignment principle. To address this, we conducted additional experiments examining threshold values $T \\in \\{0.2, 0.5, 0.8, 1.0\\}$ on the Cylinder Flow dataset. The results below demonstrate that FLARE with $T = 0$ achieves the best performance and the easiest one to implement and start with, requiring no hyperparameter tuning and no ranking/sorting operations. For positive thresholds, although the edge selection becomes more restrictive based on both velocity magnitude and alignment strength and reduce the number of added edges and computational overhead, they degrade the performance of FLARE with $T = 0$. Compared with the baseline with RMSE of 40.35, we observe stable performance gain across different $T$ values, confirming the contribution of the flow alignment principle.
>
> **Q5 & W3: Ablations Isolating Design Factors and Justifying 2-hop Locality**
>
> To isolate each factor's contribution, we conducted comprehensive ablation studies:
>
> | Method                  | $T = 0$          | $T = 0.2$        | $T = 0.5$        | $T = 0.8$        | $T = 1.0$        |
> |:------------------------------|:-----------------------------:|:-------------------------------:|:---------------------------------:|:---------------------------------:|:-------------------------------:|
> | **FLARE (ours)**           | **23.38 ± 2.50**    | -                | -                | -                | -                |
> | Ablation Threshold     | -                | 28.94 ± 3.36    | 31.55 ± 2.98    | 29.91 ± 2.61    | 30.56 ± 3.05    |
> | Ablation Multi-hop     | -                | 32.88 ± 3.06    | 29.06 ± 2.86    | 39.84 ± 3.83    | 25.40 ± 1.96    |
> | Ablation Unidirectional| -                | 34.68 ± 3.25    | 38.80 ± 3.14    | 36.23 ± 3.50    | 37.80 ± 3.50    |
>
> *Selection criterion* (Ablation Threshold): This variant tests different threshold values $T$ while keeping locality (2-hop) and directionality (flow-aligned) constant. The results show that $T = 0$ achieves the best performance, validating our choice while demonstrating reasonable robustness across different thresholds.
>
> *Locality* (Ablation Multi-hop): This variant addresses the reviewer's concern about 2-hop being optimal. We use threshold $T$ to select fewer 2-hop edges, then compensate by adding top-scoring 3-hop edges to maintain the same total edge count as FLARE. The performance degradation demonstrates that 2-hop connections are more effective than longer-range connections. This provides direct experimental evidence that 2-hop locality is effective, longer hops can hardly replace closer connections.
>
> *Directionality and flow alignment* (Ablation Unidirectional): To test whether FLARE benefits from flow alignment specifically or merely from adding more directed edges, we construct this variant separately for each threshold $T \\in \\{0.2, 0.5, 0.8, 1.0\\}$. For a given $T$, we start from all 2-hop candidates, apply FLARE's alignment rule at that threshold, and then exclude the 2-hop edges that the corresponding Threshold variant would select. From the remaining 2-hop edges, we select unidirectional edges without enforcing the flow-alignment rule; if this still yields fewer edges than FLARE, we additionally include 3-hop edges until the total number of added edges matches FLARE's edge budget. The performance degradation indicates that FLARE's gains primarily come from its specific flow-aligned selection rather than simply from adding more unidirectional connections.
>
> We have updated these ablation studies into the revised manuscript (Appendix A.2) with detailed methodology and analysis.

---

> ### Author Response · Authors · 2025-11-24
> **Response to Reviewer PxA5 (3/5)**
>
> **Q2. More evidence is needed to support the claim that PIORF "does not align with physical principles" (line 237, footnote 2). The differences between FLARE and PIORF need to be discussed more clearly, and the velocity gradient (strain rate) aspect of PIORF should be addressed. Can you explain more precisely which aspects of PIORF are less physically straightforward?**
>
> PIORF's long distance and bidirectional connections (see Fig. 1a) do not align with the following two physics principles. 1) In fluid flow, physics quantities, e.g. mass, travel first to neighbour region and then travel to distanced regions. The PIORF's long-distance rewiring does not follow this principle. 2) The net quantities are moved from single direction based on flow direction. The PIORF's long-distance rewiring does not follow this principle. FLARE, whose connections are local, directional and aligning with flow direction, is designed based on these two principles.
>
> ---
>
> **Q3. Why does FLARE underperform PIORF on Airfoil density with BSMS-GNN?**
>
> We thank the reviewer for pointing out this specific case. The main cause of FLARE's underperformance on Airfoil's density prediction with BSMS-GNN is that:
>
> FLARE's rewiring was mainly guided by velocity alignment, which is expected to be well-suited for velocity prediction but less effective for density prediction. In the compressible Airfoil dataset, density and velocity fields are coupled, so velocity-based edge selection might not optimally capture the connectivity beneficial for density prediction. To overcome this limitation, we extended FLARE to handle density field prediction by introducing separate edge sets for density field while maintaining shared node representations between the two fields (density and velocity) to account for their physical coupling. Doing so enables a density guided (2-hop based) rewiring for density prediction. With the velocity and density rewired graphs, the extended FLARE has shown improvements in both velocity and density predictions compared to the previous single velocity-guided approach. The results are given in the table below. The detailed methodology and experimental results are provided in Appendix A.4 and the updated Table.2 of our revised manuscript. We appreciate the reviewer bringing this to our attention, which leads to a meaningful improvement of our method.
>
> | Model           | Method                        | Velocity RMSE | Velocity Improv. | Density RMSE (×10³) | Density Improv. |
> |:----------------|:------------------------------|:-------------:|:----------------:|:-------------------:|:---------------:|
> | **MGN**         | Baseline                      | 35.45         | -                | 94.39               | -               |
> |                 | PIORF                         | 33.66         | 5.1%             | 95.04               | -0.7%           |
> |                 | 2-HOP-ALL                     | 34.03         | 4.0%             | 86.49               | 8.4%            |
> |                 | 2-HOP-RANDOM                  | 33.33         | 6.0%             | 95.16               | -0.8%           |
> |                 | FLARE (ours)                  | 33.27         | 6.1%             | 90.58               | 4.0%            |
> |                 | FLARE + 10% Density 2-HOP     | **31.93**     | **9.9%**         | **85.66**           | **9.2%**        |
> | **BSMS-GNN**    | Baseline                      | 46.57         | -                | 126.78              | -               |
> |                 | PIORF                         | 44.25         | 5.0%             | 99.75               | 21.3%           |
> |                 | 2-HOP-ALL                     | 45.91         | 1.4%             | 128.96              | -1.7%           |
> |                 | 2-HOP-RANDOM                  | 51.84         | -11.3%           | 195.92              | -54.6%          |
> |                 | FLARE (ours)                  | 43.63         | 6.3%             | 110.85              | 12.6%           |
> |                 | FLARE + 10% Density 2-HOP     | **39.46**     | **15.3%**        | **95.58**           | **24.6%**       |
> | **Transolver++**| Baseline                      | 40.27         | -                | 73.76               | -               |
> |                 | PIORF                         | 38.16         | 5.2%             | 73.89               | -0.2%           |
> |                 | 2-HOP-ALL                     | 38.49         | 4.4%             | 74.66               | -1.2%           |
> |                 | 2-HOP-RANDOM                  | 37.36         | 7.2%             | 73.89               | -0.2%           |
> |                 | FLARE (ours)                  | 35.40         | 12.1%            | 67.93               | 7.9%            |
> |                 | FLARE + 10% Density 2-HOP     | **34.26**     | **14.9%**        | **64.76**           | **12.2%**       |

---

> ### Author Response · Authors · 2025-11-24
> **Reponse to Reviewer PxA5 (4/5)**
>
> **Q4. Please provide a detailed computational cost comparison between baseline, PIORF, 2-HOP-ALL, and FLARE for each dataset. How much overhead does dynamic rewiring add?**
>
> We conducted comprehensive computational cost analyses on the Cylinder Flow dataset with the same computational resources mentioned in the manuscript. Our analysis covers both aggregate training/inference costs and fine-grained per-timestep dynamic rewiring overhead.
>
> ---
>
> **Aggregate Training and Inference Cost**
>
> We measured training and inference time as well as memory consumption, averaged over 50 iterations. Tables 1 and 2 compare baseline MGN, PIORF, 2-HOP-ALL, 2-HOP-RANDOM and FLARE.
>
> **Timing details:** All 2-hop rewiring methods, including FLARE, compute 2-hop connections and perform edge selection on-the-fly during training without precomputation. In contrast, PIORF precomputes the Ollivier-Ricci Curvature (ORC) ranking offline which takes around 40 minutes for the Cylinder Flow training set, is not reflected in Table 1.
>
> **Table 1: Training and Inference Time Comparison**
>
> | Model              | Training (ms)    | Inference (ms)   | Train vs MGN   | Infer vs MGN   |
> |:-----------------------|:---------------------:|:---------------------:|:-------------------:|:-------------------:|
> | MGN (Baseline)     | 52.47 ± 8.76    | 30.44 ± 7.74    | 1.000x        | 1.000x        |
> | PIORF (3% ORC)     | 58.65 ± 9.16    | 36.38 ± 9.61    | 1.118x        | 1.195x        |
> | 2-HOP-ALL          | 97.84 ± 7.14    | 33.56 ± 8.51    | 1.865x        | 1.103x        |
> | 2-HOP-RANDOM       | 58.06 ± 9.21    | 32.10 ± 7.90    | 1.107x        | 1.054x        |
> | FLARE (ours)       | 67.79 ± 8.33    | 39.54 ± 9.51    | 1.292x        | 1.299x        |
>
> **Table 2: Edge Addition and Memory Usage Comparison**
>
> | Model              | Params      | Base Edges   | Added Edges   | Total Edges   | Peak Memory (GB)   | vs MGN    |
> |:------------------------|:------------------------:|:-------------------:|:--------------------:|:-------------------:|:-----------------------:|:-----------------:|
> | MGN (Baseline)     | 2,332,930   | 10,488      | 0            | 10,488       | 0.67              | 1.00x     |
> | PIORF (3% ORC)     | 2,332,933   | 10,488      | 109          | 10,597       | 0.68              | 1.02x     |
> | 2-HOP-ALL          | 2,332,930   | 10,488      | 20,378       | 30,866       | 1.84              | 2.76x     |
> | 2-HOP-RANDOM       | 2,332,930   | 10,488      | 10,928       | 21,416       | 1.27              | 1.90x     |
> | FLARE (ours)       | 2,332,930   | 10,488      | 10,928       | 21,416       | 1.27              | 1.90x     |
>
> ---
>
> **Per-Timestep Dynamic Rewiring Overhead**
>
> To specifically quantify the dynamic rewiring overhead, we performed detailed per-timestep profiling across 30,000 measurements.
>
> **Overhead breakdown:** FLARE's dynamic graph rewiring adds 5.36 $±$ 0.23 ms per timestep, consisting of:
>
> - 2-hop edge computation: 1.98 ms (~37% of rewiring time)
> - Velocity-based edge selection: 3.39 ms (~63% of rewiring time)
>
> **Total runtime:** FLARE's complete per-timestep computation (including forward pass) is 23.18 ms (baseline MGN takes 17.82 ms). This overhead is stable and predictable, with median 5.38 ms and 95% of measurements falling within [5.18, 5.80] ms.
>
> **Time-storage tradeoff:** The 2-hop edge computation component (1.98 ms) could be converted to storage cost through precomputation and caching, like PIORF's approach. This would reduce per-timestep overhead to approximately 3.39 ms (~19% of baseline) at the expense of additional disk storage.
>
> ---
>
> **Summary**
>
> FLARE demonstrates reasonable computational overhead compared to the baseline:
>
> - Training overhead: 1.29x
> - Inference overhead: 1.30x
> - Memory overhead: 1.90x
> - Dynamic rewiring overhead: 30.1% per timestep, stable and predictable
>
> In summary, PIORF shows advantages in online computational efficiency which stems from its design choice of adding a small number of edges related to the number of nodes. Our FLARE takes a different architectural approach by adding aligned 2-hop edges, providing richer connectivity to capture flow alignment patterns. This design choice avoids dataset-specific preprocessing and offers flexibility for dynamic graphs or multi-dataset workflows.
>
> The overhead from FLARE is predictable and stable across timesteps. Given the **27.40%** improvement over PIORF and **26.03%** improvement over 2-HOP-ALL of the performance gains in our main results, we believe FLARE's cost is worthwhile considering developing computational power.

---

> ### Author Response · Authors · 2025-11-24
> **Response to Reviewer PxA5 (5/5)**
>
> **Q6. What are the detailed settings for 3-hop and 4-hop in the ablation study?**
>
> For FLARE 3-HOP and FLARE 4-HOP, instead of applying the fixed threshold $T$, we selected the same number of edges as FLARE did, but from top scored edges in 3-hop connections and 4-hop connections respectively.

---

> ### Author Response · Authors · 2025-11-28
> **Gentle Reminder**
>
> Dear Reviewer PxA5,
>
> We hope you are doing well. We are writing to kindly follow up on our previous response and to ask whether you have had a chance to review it.
>
> If you could share your feedback or confirmation at your earliest convenience, we would greatly appreciate it.
>
> Thank you again for your time and consideration.
>
> Best regards,
> Authors of the Submission 15260

---

> ### Author Response · Authors · 2025-12-02
> **Additional ablations with the original flow alignment score formulation**
>
> Following Reviewer UERp's feedback on dimensionality and velocity magnitude, we adopted the distance-normalized score $norm\\_s\_{ij}$ for all ablations with $T > 0$ in the previous response. To validate this revised formulation, we additionally repeated the ablation experiments with the original, unrevised score $s\_{ij} = \\mathbf{v}\_i^T \\mathbf{d}\_{ij}$ under comparable levels of rewiring to directly compare their performances. These additional ablation experiments are summarized below and included in Appendix A.2 of our revised manuscript.
>
> For each threshold $T \in \\{0.2, 0.5, 0.8, 1.0\\}$ used with the revised score $norm\\_s\_{ij}$, we chose a corresponding $\hat T(T)$ for $s\_{ij}$ so that, on average over the training set, the rule $s\_{ij} > \hat T(T)$ keeps a similar proportion of 2-hop candidates as $norm\\_s\_{ij} > T$.
>
> **Table 1:** Mapping from normalized thresholds $T$ to original-score thresholds $\hat T(T)$ on *CylinderFlow*.
>
> | $T$ | $\hat T(T)$ |
> |:----------------------|:-----------------------:|
> | 0.2 | 0.0014 |
> | 0.5 | 0.0036 |
> | 0.8 | 0.0058 |
> | 1.0 | 0.0079 |
>
> Under these $\hat T(T)$, we re-ran the Threshold, Multi-hop, and Unidirectional ablations with $s_{ij}$ on *CylinderFlow* (MGN backbone). The rollout RMSEs (full horizon, $\times 10^3$, mean ± SE) are given below.
>
> **Table 2:** Full-rollout RMSE on *CylinderFlow* for ablation variants using $s\_{ij}$ (values scaled by $\times 10^3$).
>
> | Method ($s_{ij}$) | $\hat T(0.2)$ | $\hat T(0.5)$ | $\hat T(0.8)$ | $\hat T(1.0)$ |
> |:-----------------------------------|:------------------------------------:|:------------------------------------:|:------------------------------------:|:------------------------------------:|
> | Ablation Threshold | 32.14 ± 3.16 | 34.40 ± 2.83 | 36.66 ± 1.94 | 36.52 ± 3.32 |
> | Ablation Multi-hop | 36.53 ± 3.07 | 33.83 ± 3.11 | 28.02 ± 2.52 | 40.54 ± 3.95 |
> | Ablation Unidirectional | 38.19 ± 2.99 | 52.09 ± 3.09 | 39.13 ± 3.11 | 36.74 ± 2.94 |
>
> The experiments with $s\_{ij}$ further support the conclusions drawn from the previous $norm\\_s\_{ij}$ ablations, while validating that the revised score $norm\\_s\_{ij}$ is the better choice. Under comparable levels of rewiring, $norm\\_s\_{ij}$ consistently achieves lower rollout error and works as intended by prioritizing edges with strong, well-aligned velocities while penalizing large $\\|\\mathbf{d}\_{ij}\\|$, which helps discourage long jumps that tend to hurt performance when higher-hop candidates are present. We adopt $norm\\_s\_{ij}$ as our flow-alignment score for $T > 0$ as claimed in Appendix A.2 of our revised manuscript.

---

### Official Review · Reviewer_UERp · 2025-11-01

**Soundness:** 3
**Presentation:** 3
**Contribution:** 2
**Rating:** 6
**Confidence:** 4

**Summary:**

This paper introduces FLARE (Flow Alignment Rewiring) for improving GNN-based CFD simulations.
Classical GNNs operating on CFD meshes suffer from limited information propagation (over-squashing) and physics-misaligned connectivity, since message passing is confined to mesh adjacency that is unrelated to flow direction.
The proposed method rewires the mesh dynamically based on local flow alignment:
* only 2-hop local connections are considered (for locality);
* edges are directional (for unidirectional transport);
* new edges are added only when the velocity aligns with the added edge.

This yields a direction-aware, physics-consistent connectivity pattern that adapts during rollout as the predicted velocity field changes.
Experiments on three datasets—CylinderFlow, Airfoil, and Tandem-Airfoil-Cruise and across three backbone architectures (MeshGraphNet, BSMS-GNN, Transolver+) show that FLARE outperforms both the prior physics-informed rewiring method PIORF and 2-hop variants.

**Strengths:**

* the proposed approach is based on the first principles of fluid mechanics—locality, directionality, and flow alignment
* model-agnostic rewiring approach that can plug into any message-passing GNN or hybrid GNN-Transformer architecture without altering its core equations
* during rollout, rewiring is based on predicted velocities, so it's dynamically adjusted
* ablations provided: direction reversal, hop distance
* clear answers to guiding questions: The experiments directly address the three motivating questions—confirming that (1) physics-informed rewiring is beneficial, (2) local directional links suffice, and (3) dynamic flow-based updates matter.

**Weaknesses:**

* the proposed rewiring way doesn't depend on fluid velocity magnitude
* the paper is mainly empirical and proposes mostly an engineering solution
* dynamic rewiring at each step may add runtime cost, but no timing or complexity study is reported
* limited physical validation metrics: The study reports RMSE/MSE only, some physics-informed metrics would improve the results

**Questions:**

* What are computational expenses for your model?
* Can your approach be extended to multiphase flows or multi-physics simulations?
* How large is the runtime overhead of recomputing the rewired graph at each timestep?
* Your alignment score is dimensional, so how will you handle the cases of extremely slow flows in some points or extremely strong flows?
* Could the same idea be applied beyond fluids (e.g., heat diffusion, elasticity)?

---

> ### Author Response · Authors · 2025-11-24
> **Response to Reviewer UERp (1/4)**
>
> Dear Reviewer UERp,
>
> We sincerely appreciate your thoughtful review and your recognition that our rewiring scheme is grounded in first-principles fluid mechanics (locality, directionality, and flow alignment), remains model-agnostic across GNN and hybrid GNN--Transformer architectures, and that our dynamic velocity-based rewiring and ablations clearly address the guiding questions. We address your comments point by point below.
>
> ---
> **Q1. What are computational expenses for your model?**
>
> Thank you for raising this important question regarding computational expenses.
>
> We conducted an analysis of training and inference time, as well as memory consumption, averaged over 50 iterations on the Cylinder Flow dataset utilizing the same computational resources as claimed in the paper. Tables 1 and 2 report per-step time and memory expense, together with the number of edges added.
>
> **Timing Details:** For all 2-hop rewiring methods, including FLARE, we compute 2-hop connections and perform edge selection during training without precomputation for saving storage. In contrast, PIORF has to precompute the Ollivier-Ricci Curvature (ORC) ranking offline, and the ORC ranking computation itself is time-consuming (about 40 minutes for Cylinder Flow training set) which is not reflected in the reported training time in Table 1.
>
> **Table 1: Training and Inference Time Comparison**
>
> | Model | Training (ms) | Inference (ms) | Train vs MGN | Infer vs MGN |
> |:----------|:-----------------:|:-------------------:|:-----------------:|:-----------------:|
> | MGN (Baseline) | 52.47 ± 8.76 | 30.44 ± 7.74 | 1.000x | 1.000x |
> | PIORF (3% ORC) | 58.65 ± 9.16 | 36.38 ± 9.61 | 1.118x | 1.195x |
> | 2-HOP-ALL | 97.84 ± 7.14 | 33.56 ± 8.51 | 1.865x | 1.103x |
> | 2-HOP-RANDOM | 58.06 ± 9.21 | 32.10 ± 7.90 | 1.107x | 1.054x |
> | FLARE (ours) | 67.79 ± 8.33 | 39.54 ± 9.51 | 1.292x | 1.299x |
>
>
> **Table 2: Edge Addition and Memory Usage Comparison**
>
> | Model | Params | Base Edges | Added Edges | Total Edges | Peak Memory (GB) | vs MGN |
> |:--------------|:--------------:|:---------------:|:----------------:|:----------------:|:-----------------------:|:-------------:|
> | MGN (Baseline) | 2,332,930 | 10,488 | 0 | 10,488 | 0.67 | 1.00x |
> | PIORF (3% ORC) | 2,332,933 | 10,488 | 109 | 10,597 | 0.68 | 1.02x |
> | 2-HOP-ALL | 2,332,930 | 10,488 | 20,378 | 30,866 | 1.84 | 2.76x |
> | 2-HOP-RANDOM | 2,332,930 | 10,488 | 10,928 | 21,416 | 1.27 | 1.90x |
> | FLARE (ours) | 2,332,930 | 10,488 | 10,928 | 21,416 | 1.27 | 1.90x |
>
> **Summary**
>
> FLARE incurs moderate overhead compared to PIORF: 1.29x training time, 1.30x inference time, and 1.90x memory usage. Considering FLARE achieves up to **27.40%** improvement over PIORF and **26.03%** improvement over 2-HOP-ALL of the performance gains in our main results, we believe that FLARE offers a worthy trade-off from computational expenses to performance.
>
> ---

---

> ### Author Response · Authors · 2025-11-24
> **Response to Reviewer UERp (2/4)**
>
> **Q2. Can your approach be extended to multiphase flows or multi-physics simulations?**
>
> Thank you for your valuable question about extending FLARE to more complex physical scenarios.
>
> **Multi-physics simulations:** We believe that FLARE, which was primarily designed to improve velocity prediction through velocity-guided rewiring, can be extended to multi-physics simulations by introducing multiple rewired graphs, one for velocity prediction and the rest for other variable predictions. To demonstrate this idea, we have implemented it to the same compressible Airfoil dataset as the original manuscript, where density is a transported variable. Since mass transport is local and depends on density difference between two local regions, we can retain the 2-hop connections and select to $k$% 2-hop edges with the highest density difference between two 2-hop nodes. Note that, however, the connections should be bidirectional for mass transport. The rewiring scheme for density is given below:
>
> For each node pair $(i, j)$ connected by a 2-hop path, we compute $\Delta\rho_{ij} = |\rho_i - \rho_j|$, where $\rho_i$ and $\rho_j$ are the density values at nodes $i$ and $j$. We then rank all 2-hop node pairs by $\Delta\rho_{ij}$ and select the top 10% edges with the highest $\Delta\rho_{ij}$ to form the density rewiring edge set $\mathcal{E}_{\text{density}}$. Unlike FLARE rewiring where edges are mainly directional (aligned with flow direction), density edges are bidirectional to represent local mass transport between regions. That is, if $(i,j)$ is selected based on density, we add both $(i,j)$ and $(j,i)$ to the rewired graph for density prediction.
>
> Once the rewired graphs for velocity and density are constructed, they are used together to improve the density prediction. In each layer of the network, each graph is used to update the nodes and edges features independently. Then, a MLP takes the features from the two graphs and determines the final features for the next layer. The table below shows that the extended FLARE is indeed effective for both velocity and density.
>
> | Model | Method | Velocity RMSE | Velocity Improv. | Density RMSE (×10³) | Density Improv. |
> |:--------|:-----|-----------:|--------------:|-----------------:|-------------:|
> | **MGN** | Baseline | 35.45 | - | 94.39 | - |
> | | PIORF | 33.66 | 5.1% | 95.04 | -0.7% |
> | | 2-HOP-ALL | 34.03 | 4.0% | 86.49 | 8.4% |
> | | 2-HOP-RANDOM | 33.33 | 6.0% | 95.16 | -0.8% |
> | | FLARE (ours) | 33.27 | 6.1% | 90.58 | 4.0% |
> | | FLARE + 10% Density 2-HOP | **31.93** | **9.9%** | **85.66** | **9.2%** |
> | **BSMS-GNN** | Baseline | 46.57 | - | 126.78 | - |
> | | PIORF | 44.25 | 5.0% | 99.75 | 21.3% |
> | | 2-HOP-ALL | 45.91 | 1.4% | 128.96 | -1.7% |
> | | 2-HOP-RANDOM | 51.84 | -11.3% | 195.92 | -54.6% |
> | | FLARE (ours) | 43.63 | 6.3% | 110.85 | 12.6% |
> | | FLARE + 10% Density 2-HOP | **39.46** | **15.3%** | **95.58** | **24.6%** |
> | **Transolver++** | Baseline | 40.27 | - | 73.76 | - |
> | | PIORF | 38.16 | 5.2% | 73.89 | -0.2% |
> | | 2-HOP-ALL | 38.49 | 4.4% | 74.66 | -1.2% |
> | | 2-HOP-RANDOM | 37.36 | 7.2% | 73.89 | -0.2% |
> | | FLARE (ours) | 35.40 | 12.1% | 67.93 | 7.9% |
> | | FLARE + 10% Density 2-HOP | **34.26** | **14.9%** | **64.76** | **12.2%** |
>
> Similarly, we can generate rewired graphs for other transported variables, particularly temperature, thus constituting an extended FLARE for thermal or reacting flows, which are classical multi-physics flow problems. We will evaluate this idea for multi-physics simulations in the coming future. We would like to highlight that many existing works, including PIORF, examine their methods on single physics simulations.
>
> **Multiphase Flows:** FLARE can be beneficial to multiphase simulations but is likely more applicable for scenarios involving only fluids (i.e., liquid and gas phases), such as free-surface modelling. The reason is that, for such problems, the typical approaches are level-set and volume-of-fluid methods, both of which involve solving (an) additional transport equation(s) of notional scalar(s) (i.e., a level-set function and multiple fraction functions, respectively), for which we may generate rewired graph(s). In contrast, the same cannot be said for computational structural dynamics, so we are currently not confident that FLARE will improve predictions of fluid-structure interactions.
>
> Hence, FLARE is conceptually possible to be extended to liquid-gas interactions, which is a specific type of multiphase cases. The promising results from our current extended FLARE for density and velocity suggests that such implementation should improve predictions of liquid-gas interactions. However, to perform such experiments, we need to set up a new dataset and revise our codes for it, which is too ambitious to complete over the rebuttal period. Hence, we will evaluate this idea for multiphase simulations in the future, noting that most existing works in the physics-informed AI community consider only single-phase problems.

---

> ### Author Response · Authors · 2025-11-24
> **Response to Reviewer UERp (3/4)**
>
> **Q3. How large is the runtime overhead of recomputing the rewired graph at each timestep?**
>
> We conducted runtime analysis experiment of FLARE's dynamic graph rewiring on the Cylinder Flow dataset (with the same computational resources as mentioned in the manuscript). Computing 2-hop edges takes 1.98 ms and edge selection takes 3.39 ms on average per frame. FLARE's total runtime, including the forward pass is 23.18 ms per timestep on average. We measured this across 30k timesteps and found the overhead is consistent, with a median of 5.38 ms and 95% of measurements between 5.18 and 5.80 ms.
>
> Meanwhile, the time cost of 2-hop edge computation could be transferred to storage cost if we precompute and cache the connectivity, similar to how PIORF handles preprocessing. This would cut the per-timestep overhead down to about 3.39ms (about 19% of baseline) while requiring extra disk storage.
>
> The runtime overhead is predictable and stable, and we believe this 5.38ms overhead is worthwhile considering the performance gain and raising limits of modern hardware.
>
> ---
>
> **Q4. Your alignment score is dimensional, so how will you handle the cases of extremely slow flows in some points or extremely strong flows?**
>
> Thank you for this important question. We would like to clarify that using raw velocity in our score computation helps us handle flows with varying magnitudes effectively when the threshold $T > 0$.
>
> We have modified our flow alignment score formulation to better handle spatial scale variations. Previously, we computed the score as
> $s\_{ij} = \\mathbf{v}\_i^T \\mathbf{d}\_{ij}$, where $\\mathbf{v}\_i$ is the velocity at the sender node and $\\mathbf{d}\_{ij} = \\mathbf{x}\_j - \\mathbf{x}\_i$ is the displacement vector. We now incorporate distance-dependent normalization:
> $norm\\_s\_{ij} = \\mathbf{v}\_i^T \\dfrac{\\mathbf{d}\_{ij}}{\\lVert \\mathbf{d}\_{ij} \\rVert \\cdot \\lVert \\mathbf{d}\_{ij} \\rVert^{0.5}}$ for $T > 0$. Importantly, we use raw velocity vectors $\\mathbf{v}\_i$ without normalizing them to unit length. This means the score magnitude naturally reflects both the velocity magnitude and the flow alignment, which is physically meaningful: regions with stronger flows produce higher scores and are prioritized for edge addition, while slower flow regions contribute relatively less. The distance-dependent normalization could provide better balance across different spatial scales. When the threshold $T$ is set above 0, the edge selection explicitly accounts for velocity magnitude through the dimensional score.
>
> To validate our design choices, we conducted ablation studies on the Cylinder Flow dataset with $T \\in \\{0.2, 0.5, 0.8, 1.0\\}$.
>
>
> | Method | $T=0$ | $T=0.2$ | $T=0.5$ | $T=0.8$ | $T=1.0$ |
> |:---------------|:---------------------------:|:-----------------------------:|:------------------------------:|:------------------------------:|:------------------------------:|
> | **FLARE (ours)** | **23.38 ± 2.50** | - | - | - | - |
> | Ablation Threshold | - | 28.94 ± 3.36 | 31.55 ± 2.98 | 29.91 ± 2.61 | 30.56 ± 3.05 |
> | Ablation Multihop | - | 32.88 ± 3.06 | 29.06 ± 2.86 | 39.84 ± 3.83 | 25.40 ± 1.96 |
> | Ablation Unidirectional | - | 34.68 ± 3.25 | 38.80 ± 3.14 | 36.23 ± 3.50 | 37.80 ± 3.50 |
>
> **Ablation Threshold:** Select only 2-hop edges with $norm\\_s\_{ij} > T$, reducing the number of selected edges as $T$ increases. This ablation demonstrates threshold sensitivity and validates our choice of $T=0$ as an effective setting.
>
> **Ablation Multi-hop:** Apply threshold $T$ for 2-hop selection, then add top-scoring 3-hop edges to maintain the same total edge count as FLARE (where $T=0$). This ablation tests the importance of 2-hop locality that whether fewer high-quality 2-hop edges plus longer-range 3-hop connections can match FLARE's performance. The degraded performance confirms that 2-hop locality is crucial.
>
> **Ablation Unidirectional:** Exclude all 2-hop edges Ablation Threshold would select (under different $T$) and instead select strictly unidirectional edges from the rest of 2-hop candidates, plus 3-hop edges to match FLARE's total edge count, to test the benefits of flow alignment versus unidirectional edges. This ablation isolates the contribution of flow alignment versus mere directionality, demonstrating that FLARE's benefit comes from physical flow alignment, not simply from adding unidirectional edges.
>
> The results show that FLARE with $T=0$ consistently outperforms all ablation variants, which is also the easiest to implement configuration, requiring no threshold tuning and no sorting/ranking operations while achieving the best performance. The ablations validate our key design choices: 2-hop locality is crucial and can hardly be replaced by longer-range connections (Ablation Multi-hop), and the benefits indeed come from flow alignment specifically, not just directionality (Ablation Unidirectional).
>
> We have included these ablation experiments in our revised manuscript (Appendix A.2).

---

> ### Author Response · Authors · 2025-11-24
> **Response to Reviewer UERp (4/4)**
>
> **Q5. Could the same idea be applied beyond fluids (e.g., heat diffusion, elasticity)?**
>
> Thank you for your question regarding exploration beyond fluids.
>
> By "beyond fluids", we guess the reviewer is asking if the same idea can be applied to solid state, the only phase that is not considered as a fluid. In this case, the applicability of FLARE is dependent on the problem. For the two examples given by the reviewer, namely heat diffusion and elasticity, we believe that only the former will benefit from FLARE since heat diffusion within solid is in general governed by a parabolic partial differential heat equation, which consists of a transported temperature variable. Hence, we can generate a rewired graph for the temperature component predictions by retaining the 2-hop connections and selecting top $k$% 2-hop edges with the highest thermal gradients between two nodes. Note that the connections should be bidirectional for heat exchange. We have implemented this idea as an extended FLARE for improving simulation of density for the same compressible Airfoil dataset as the original manuscript. The results given in the table in the previous response show that this extension is effective for both velocity and density. We endeavour to explore this idea on heat diffusion in the coming future.
>
> On the other hand, we are not confident that FLARE will improve structural dynamics problems, such as elasticity, since, unlike fluids, they are typically not governed by transport equation. Such a limitation is reasonable since FLARE is design based on principle of fluid physics after all, but we are open to developing models for computational structural dynamics and welcome any collaborations with experts from the structural mechanics community.

---

> ### Author Response · Authors · 2025-11-28
> **Gentle Reminder**
>
> Dear Reviewer UERp,
>
> We hope this message finds you well. We are writing to follow up on our previous response to your comments and to see whether you have had an opportunity to review it.
>
> We truly appreciate the feedback you have already provided, and if you could share any further thoughts or a brief confirmation when convenient, it would be very helpful.
>
> Thank you again for your time and support.
>
> Sincerely,
> Authors of the Submission 15260

---

### Comment · Area_Chair_j65W · 2025-11-26
**reminder**

Dear Reviewers,

The authors have now posted their responses to your comments. As the next step in the discussion phase, please take a moment to review their rebuttal and engage with them through the discussion forum. Thank you for your continued effort and thoughtful contributions to this review.

Best,

Your AC

---

### Author Response · Authors · 2025-12-02
**Author Final Remarks**

We sincerely appreciate the Area Chair's time and effort in evaluating our paper, and we thank all four reviewers for their thoughtful and constructive feedback. We briefly summarize below the main issues addressed in our rebuttal.

Across reviews, several strengths are repeatedly emphasized. Reviewers describe FLARE as:

- "the proposed approach is based on the first principles of fluid mechanics—**locality, directionality, and flow alignment**" and a "**model-agnostic rewiring approach that can plug into any message-passing GNN or hybrid GNN-Transformer architecture without altering its core equations**" (UERp);
- a method that shows "**consistent and substantial performance improvements over PIORF and structural baselines across three diverse datasets (unsteady, steady, compressible/incompressible) and 3 architectures**" (PxA5);
- "a **lightweight, plug-and-play rule that rewires 2-hop, flow-aligned, directional edges dynamically well-matched to advective transport and slows long-horizon error growth**" (5y92);
- a "**physics-informed rewiring heuristic uses simple fluid-dynamics principles and often gives a gain in performance**" and a method that "**is easy to implement**" (ebA5).

Concretely, FLARE augments the baseline mesh with **2‑hop, flow‑aligned, directional edges** that are dynamically selected at each rollout step and can be plugged into MGN, BSMS‑GNN, and Transolver++, yielding **27.40%** and **26.03%** lower rollout velocity RMSE than PIORF and 2‑HOP‑ALL on CylinderFlow, with consistent gains on Airfoil and Tandem‑Airfoil.

---

### Methodology

Taking reviewer UERp's feedback on dimensionality and velocity magnitude, we revised the flow-alignment score with distance-normalization for new ablations with $T>0$, the original and revised score select the same edges for $T=0$. With threshold, multi-hop, and unidirectional ablations, we show that (i) **2-hop locality is crucial** and can hardly be replaced by longer-range hops, (ii) **performance gains come from true flow alignment** rather than simply adding unidirectional edges, and (iii) **$T=0$ is a simple, robust default** that requires no extra tuning. The **velocity‑turning‑angle analysis** confirms that FLARE reduces error across dynamical regimes and is **most beneficial in highly dynamic regions**. We also extended FLARE to the compressible *Airfoil* dataset via a **density‑guided 2‑hop rewiring** combined with the velocity‑guided graph, which achieves the **best velocity and density accuracy**, with velocity RMSE reductions of **9.9%, 15.3%, 14.9%** and density reductions of **9.2%, 24.6%, 12.2%** for MGN, BSMS‑GNN, and Transolver++. We further discussed how the same construction can apply to other transported scalars (e.g., temperature).

---

### Computational Cost

In response to UERp, PxA5, and ebA5, we ran a detailed **runtime / memory cost study and per-timestep profiling** on CylinderFlow. FLARE shows a **moderate, predictable overhead** relative to baselines, and profiling over 30k steps indicates that **dynamic rewiring accounts for only a small, stable fraction of the step time**. Together with the accuracy gains above, this indicates a favourable cost–accuracy trade‑off for rollout prediction.

---

### Physics Clarifications

We tied FLARE more explicitly to CFD practice and addressed the conceptual questions raised by the reviewers. For **time-stepping and locality**, we recalled the standard CFL condition to explain that in CFD a fluid parcel moves less than one cell per step, so one hop already exceeds the physical displacement; accurate simulation therefore relies on multiple numerical updates (or layers), and FLARE operates in this regime by keeping the same time step as the baselines while enriching local connectivity rather than modifying the temporal discretization. For **backflow and vortices**, we clarified that FLARE does not force one-way transport, the alignment rule is applied independently in each direction, reporting **≈150.8 bidirectional edge pairs** among **≈9.6k** directed edges per frame on CylinderFlow. Finally, we addressed **conservation and PIORF-related questions** by (i) explaining that our models are trained against CFD solutions that already satisfy the conservation laws, (ii) clarifying that PIORF's long-range, bidirectional edges are best viewed as a generic non-local message-passing mechanism rather than a discretization of a specific fluid operator, and (iii) providing results for a directional PIORF-Aligned variant, which remains weaker than FLARE.

---

Before the stop of the discussion, our paper received scores **6, 6, 4, 4** (from **6, 4, 4, 4**), where reviewer **ebA5** stated being "satisfied with the responses" and raised their score from 4 to 6.

We would like to once again express our sincere gratitude to the Area Chair and all reviewers for their time, effort, and constructive feedback, which has been invaluable in strengthening our work.

---

### Meta-Review · Area_Chair_4A87 · 2026-01-08

**Summary:**

They presented a graph rewiting method for fluid simulation. PIORF, a previos work on this topic, is not specialized to fluid simulation so it makes many distant edges. However, PIORF's strategy does not well aligned with the physical rules of the fluid simulation (cf. Fig. 2 in the paper). Threfore, they revisit the problem and propose to connect 2-hop neighbors. The overall idea is surprisingly simple but its main idea is timely since PIORF is the first rewiring work in mesh graph net. After that, they conduct experiments on airflow and cylinderical flow datasets.

**Reviewer Concerns:**

Reviewers raised several concerns, rangining from the increased model complecity to the motiviation fo 2-hop edges (why not longer hops). However, the most critical concers are that their experiments are too simple. One reviewer aksed about the possiblity of testing in multi-physics and/or more complicated fluid dynamics. I also fully agree on this view. This paper has only three very basic fluid dynamics. They need to consider more complicated navier-stokes experiments (with high Raynolds numbers).

Overall, I think the main idea is good and the topic is very timely. I would recommend accpetance if they conducted experiments on complicated fluid dynamics. However, I think this paper has potential and they can enhance the experiment section significantly.

**Reviewer Scores:**

One reviewer may be satisfied with the rebuttal. However, I think it seems unlcease about other reviewers. In addition to them, I personally see that their experiments are too naive althogh the idea is simple yet effective and timely.

---

### Decision · Program_Chairs · 2026-01-26

Reject